# Multiple polarity kinases inhibit phase separation of F-BAR protein Cdc15 and antagonize cytokinetic ring assembly in fission yeast

Rahul Bhattacharjee[1], Aaron R Hall[2], MariaSanta C Mangione[1†], Maya G Igarashi[1‡], Rachel H Roberts-Galbraith[1§], Jun-Song Chen[1], Dimitrios Vavylonis[2,3], Kathleen L Gould[1]*

[1]Department of Cell and Developmental Biology, Vanderbilt University School of Medicine, Nashville, United States; [2]Department of Physics, Lehigh University, Bethlehem, United States; [3]Center for Computational Biology, Flatiron Institute, New York, United States

*For correspondence:
kathy.gould@vanderbilt.edu

Present address: †Department of Internal Medicine, University of Texas Health Sciences Center, Dallas, United States; ‡University of Chicago, Chicago, IL, United States; §University of Georgia, Athens, GA, United States

Competing interest: The authors declare that no competing interests exist.

**Abstract** The F-BAR protein Cdc15 is essential for cytokinesis in *Schizosaccharomyces pombe* and plays a key role in attaching the cytokinetic ring (CR) to the plasma membrane (PM). Cdc15's abilities to bind to the membrane and oligomerize via its F-BAR domain are inhibited by phosphorylation of its intrinsically disordered region (IDR). Multiple cell polarity kinases regulate Cdc15 IDR phosphostate, and of these the DYRK kinase Pom1 phosphorylation sites on Cdc15 have been shown in vivo to prevent CR formation at cell tips. Here, we compared the ability of Pom1 to control Cdc15 phosphostate and cortical localization to that of other Cdc15 kinases: Kin1, Pck1, and Shk1. We identified distinct but overlapping cohorts of Cdc15 phosphorylation sites targeted by each kinase, and the number of sites correlated with each kinases' abilities to influence Cdc15 PM localization. Coarse-grained simulations predicted that cumulative IDR phosphorylation moves the IDRs of a dimer apart and toward the F-BAR tips. Further, simulations indicated that the overall negative charge of phosphorylation masks positively charged amino acids necessary for F-BAR oligomerization and membrane interaction. Finally, simulations suggested that dephosphorylated Cdc15 undergoes phase separation driven by IDR interactions. Indeed, dephosphorylated but not phosphorylated Cdc15 undergoes liquid–liquid phase separation to form droplets in vitro that recruit Cdc15 binding partners. In cells, Cdc15 phosphomutants also formed PM-bound condensates that recruit other CR components. Together, we propose that a threshold of Cdc15 phosphorylation by assorted kinases prevents Cdc15 condensation on the PM and antagonizes CR assembly.

## Editor's evaluation

Yeast and humans use an actin-myosin II-based division apparatus, which generates forces for cell division. A key factor that regulates cell division apparatus assembly and ensures its stability is Cdc15. The authors use a combination of biochemistry, advanced imaging, and genetics to elucidate how protein kinases, i.e., molecules that attach signaling phosphate moieties to client proteins, regulate the formation of 'condensates' of Cdc15, and how this modulates the correct assembly of the cell division apparatus.

## Introduction

Cytokinesis is the final step in the cell division cycle that is achieved in many eukaryotes by employing an actin- and myosin-based cytokinetic ring (CR). CR assembly, constriction, and disassembly are each regulated and coordinated with other cell cycle events to protect genome integrity. Throughout cell division, the CR remains attached to the plasma membrane (PM) so that CR constriction results in PM apposition, fusion, and ultimately the physical separation of two daughter cells.

In the fission yeast *Schizosaccharomyces pombe,* the CR comprises approximately 40 proteins, including PM-bound scaffolds. One of these PM-binding proteins is the essential F-BAR protein Cdc15. Cdc15 is one of the first and most abundant components detected at the CR (*Nurse et al., 1976*; *Fankhauser et al., 1995*; *Wu et al., 2003*; *Wu and Pollard, 2005*). An N-terminal F-BAR domain oligomerizes, binds membranes, and binds the formin Cdc12 and the paxillin-related Pxl1 (*Carnahan and Gould, 2003*; *McDonald et al., 2015*; *Willet et al., 2015a*; *Snider et al., 2020*; *Snider et al., 2022*). A C-terminal SH3 domain binds multiple CR components, including the C2 domain protein Fic1 and Pxl1 at a second binding site (*Roberts-Galbraith et al., 2009*; *Cortés et al., 2015*; *Ren et al., 2015*; *Martín-García et al., 2018*; *Bhattacharjee et al., 2020*). Between the F-BAR and SH3 domains is a long stretch of amino acids predicted to form an intrinsically disordered region (IDR) essential for Cdc15 function (*Mangione et al., 2019*). Reducing the amount of Cdc15, preventing Cdc15 membrane binding and/or oligomerization, or deleting the Cdc15 SH3 domain destabilizes the CR during anaphase and leads to cytokinesis failure (*Roberts-Galbraith et al., 2009*; *Arasada and Pollard, 2011*; *McDonald et al., 2015*).

Cdc15 functions are regulated by its phosphostate (*Fankhauser et al., 1995*; *Roberts-Galbraith and Gould, 2010*; *Bhattacharjee et al., 2020*). During interphase, Cdc15 is primarily cytosolic and hyperphosphorylated (≥35 phosphorylation sites). Dephosphorylation of Cdc15 at mitotic onset is accompanied by its re-localization to the medial cortex, oligomerization, and interaction with protein partners.

Three distinct polarity kinases (DYRK kinase Pom1, MARK/PAR-1 kinase Kin1, and p21-activated kinase Pak1/Shk1/Orb2) are implicated in regulating Cdc15 phosphostatus in vivo and are capable of phosphorylating Cdc15 directly in vitro (*Kettenbach et al., 2015*; *Lee et al., 2018*; *Bhattacharjee et al., 2020*; *Magliozzi et al., 2020*). The effect of Pom1 phosphorylation on Cdc15 is the best understood. Concentrated at the cortex of growing cell ends, Pom1 signals to antagonize CR formation and septation at cell tips (*Celton-Morizur et al., 2006*; *Padte et al., 2006*; *Huang et al., 2007*; *Martin and Berthelot-Grosjean, 2009*; *Hachet et al., 2011*; *Bhatia et al., 2014*) in part by phosphorylating Cdc15 and blocking Cdc15's ability to bind membrane and Pxl1 (*Bhattacharjee et al., 2020*). Considerably less is known about how Kin1 and Shk1 influence Cdc15, and it is not understood why three distinct polarity kinases would be used to modulate Cdc15 function. One possibility is that each kinase regulates different aspects of Cdc15 function since they are reported to phosphorylate largely nonoverlapping sites (*Kettenbach et al., 2015*; *Lee et al., 2018*; *Magliozzi et al., 2020*). Alternatively, the three kinases may modulate Cdc15's properties similarly but each kinase is deployed with different spatial and temporal specificity. This idea is consistent with the distinct but overlapping localizations of Pom1, Kin1, and Shk1 at cell tips and the CR (*Huang et al., 2007*; *Hachet and Simanis, 2008*; *Bhatia et al., 2014*; *Lee et al., 2018*; *Magliozzi et al., 2020*).

To dissect the importance of deploying multiple kinases to regulate Cdc15, we determined the individual contributions of Pom1, Kin1, and Shk1. We established that the cohort of sites phosphorylated by each kinase overlaps and discovered that a fourth kinase, protein kinase C Pck1, modulates Cdc15 phosphostate in vivo and phosphorylates Cdc15 directly in vitro. All four kinases affected Cdc15 membrane localization in a manner that correlated with the number of targeted phosphorylation sites. Coarse grain molecular dynamics simulations indicated that phosphorylation impacts the organization of the IDRs relative to the F-BAR domain within a dimer, with increasing phosphorylation promoting separation of the two IDRs and their clustering around F-BAR domain tips. Simulations also suggested that dephosphorylation might promote IDR-mediated liquid–liquid phase separation (LLPS). Using in vitro biochemical assays, we confirmed that IDR phosphorylation state governed Cdc15's ability to form droplets and recruit binding partners into them. In accord with in vitro studies, Cdc15 phosphomutants formed cortical condensates that recruited other CR components within cells. Overall, our data indicate that Cdc15 condenses on the membrane and scaffolds CR components, and this function is controlled by limiting Cdc15 phosphorylation with spatiotemporal precision.

## Results

### Multiple polarity kinases modulate Cdc15 phosphostatus

When examined by SDS-PAGE and immunoblotting, the phosphorylated forms of Cdc15 migrate slowly and heterogeneously to generate a broad band that can be collapsed to a single band by phosphatase treatment (*Figure 1A*). Despite the many sites Pom1 targets in vitro (*Kettenbach et al., 2015*; *Bhattacharjee et al., 2020*), we observed only a modest change in Cdc15 SDS-PAGE mobility in *pom1Δ* cells (*Bhattacharjee et al., 2020*). Kin1 also phosphorylates Cdc15 (*Kettenbach et al., 2015*; *Lee et al., 2018*), so we tested whether inactivating Kin1 alone or in combination with Pom1 resulted in a more substantial change in Cdc15 phosphostate. To do this, we used *pom1*(T778G) (*pom1$^{as1}$*) (*Padte et al., 2006*) and *kin1*(F220G) (*kin1$^{as1}$*) (*Cadou et al., 2010*; *Lee et al., 2018*) to inhibit Pom1 and Kin1 kinase activity using the ATP analog 3MB-PP1. We observed small increases in Cdc15 SDS-PAGE mobility when either kinase was individually inhibited and an additive increase when both kinases were inhibited (*Figure 1B*). However, this increase in mobility did not match the extent of dephosphorylation observed for the Pom1 site mutant, Cdc15-22A (*Figure 1B*). To rule out that inhibitor treatment of the *kin1$^{as1}$* mutant was unable to abrogate Kin1 activity, we examined Cdc15 phosphorylation state in *kin1Δ* compared to *cdc15-22A*. Again, we found modest change in Cdc15 mobility in *kin1Δ* cells (*Figure 1C*). We also found that Kin1 loss did not significantly change the migration of Cdc15-22A, which, as we showed previously, is still phosphorylated (*Figure 1C*; *Bhattacharjee et al., 2020*). These results indicate that many Cdc15 phosphosites remain modified in the absence of Pom1 and Kin1 function.

The p21-activated kinase Pak1/Shk1/Orb2 is a third polarity kinase that phosphorylates Cdc15 (*Magliozzi et al., 2020*). Thus, we examined how inhibiting Shk1 alone or in combination with Pom1 and Kin1 affected Cdc15 phosphostate. For this, we used the ATP analog 3BrB-PP1-sensitive *shk1$^{as2}$* (M460A) (*Cipak et al., 2011*; *Magliozzi et al., 2020*) or temperature-sensitive *orb2-34* alleles (*Verde et al., 1995*). We found that in both cases Shk1 inhibition modestly decreased Cdc15 phosphorylation (*Figure 1D and E*). Larger reductions in Cdc15 phosphorylation were observed when both Shk1 and Pom1, or Shk1 and Kin1 were inhibited (*Figure 1F*). A still larger decrease in Cdc15 phosphorylation was observed when all three polarity kinases were inhibited; however, Cdc15 remained phosphorylated and more phosphorylated than Pom1-resistant Cdc15-22A (*Figure 1F*). Taken together, these results indicate that still other kinases phosphorylate Cdc15.

To identify additional Cdc15 regulators, we screened all nonessential protein kinases known to participate in polarity or cell division and found that Cdc15 phosphostatus was reduced in *pck1Δ* cells (*Figure 1G*). Pck1 is one of two protein kinase C enzymes in *S. pombe*; it acts downstream of Rho1 GTPase to regulate septum deposition and participates in the cell wall integrity pathway (*Toda et al., 1993*; *Kobori et al., 1994*; *Arellano et al., 1999*; *Sánchez-Mir et al., 2014*). Cdc15 phosphorylation was also modestly reduced by inhibiting the analog 3BrB-PP1 sensitive *pck1$^{as2}$* (M744G) mutant (*Bohnert et al., 2020*; *Figure 1H*). We next combined *pck1$^{as2}$* with the other three analog-sensitive mutations. When all four kinases were simultaneously inhibited, Cdc15 SDS-mobility approached that of Cdc15-22A (*Figure 1I*). While phosphatase treatment of Cdc15-22A revealed residual phosphorylation (*Figure 1I*), it appeared that Pom1, Kin1, Shk1, and Pck1 are together responsible for the bulk of Cdc15 phosphorylation.

### Polarity kinases phosphorylate Cdc15 on shared and distinct sites

We performed in vitro assays to confirm that Cdc15 can be directly phosphorylated by Pck1 and to localize and compare the phosphorylation sites of these four protein kinases first using two fragments of Cdc15 (*Figure 2—figure supplement 1A*). Kin1 was proposed to phosphorylate Cdc15 on sites distinct from those targeted by Pom1 (*Kettenbach et al., 2015*; *Lee et al., 2018*) and concordant with this finding, Kin1 phosphorylated Cdc15C (amino acids 441–927) but not Cdc15N (amino acids 1–460), whereas Pom1 is able to phosphorylate both fragments (*Bhattacharjee et al., 2020*). Shk1 and Pck1 also phosphorylated Cdc15C but not Cdc15N (*Figure 2—figure supplement 1B*). Substituting the Pom1 phosphorylation sites in Cdc15C (*Bhattacharjee et al., 2020*) with alanines (Cdc15-19A) or aspartic acids (Cdc15-19D) did not preclude phosphorylation by Kin1, Shk1, or Pck1 (*Figure 2—figure supplement 1C*; also see Figure 3A) consistent with the idea that Shk1, Kin1, and Pck1 each phosphorylate Cdc15 at some distinct sites.

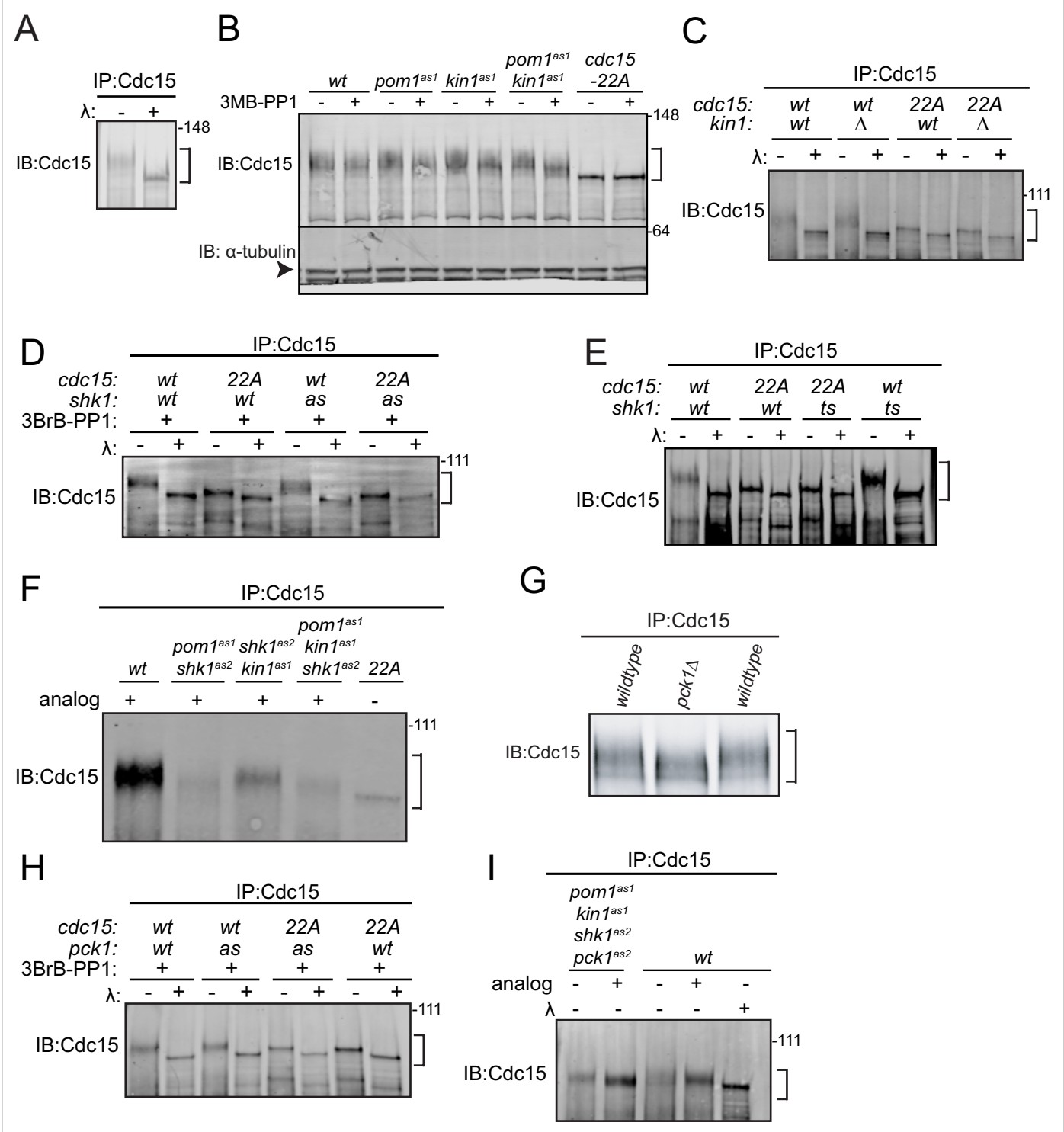

**Figure 1.** Cdc15 phosphorylation is regulated by polarity kinases Kin1, Shk1, and Pck1. Denatured protein lysates were prepared from the indicated strains. Anti-Cdc15 was used to immunoprecipitate (IP) Cdc15, which was then treated with λ protein phosphatase (λ) or buffer control. IP samples were separated by SDS-PAGE and immunoblotted for Cdc15 (**A, C, D, E, F, G, H and I**). (**B**) Denatured protein lysates from the indicated strains were separated by SDS-PAGE and then immunoblotted for Cdc15 and α-tubulin. In (**B**), α-tubulin is indicated with an arrowhead. The positions of Cdc15 isoforms are indicated with brackets. The indicated molecular mass markers are in kDa.

The online version of this article includes the following source data for figure 1:

**Source data 1.** Raw Western blot images with labeled band of interest.

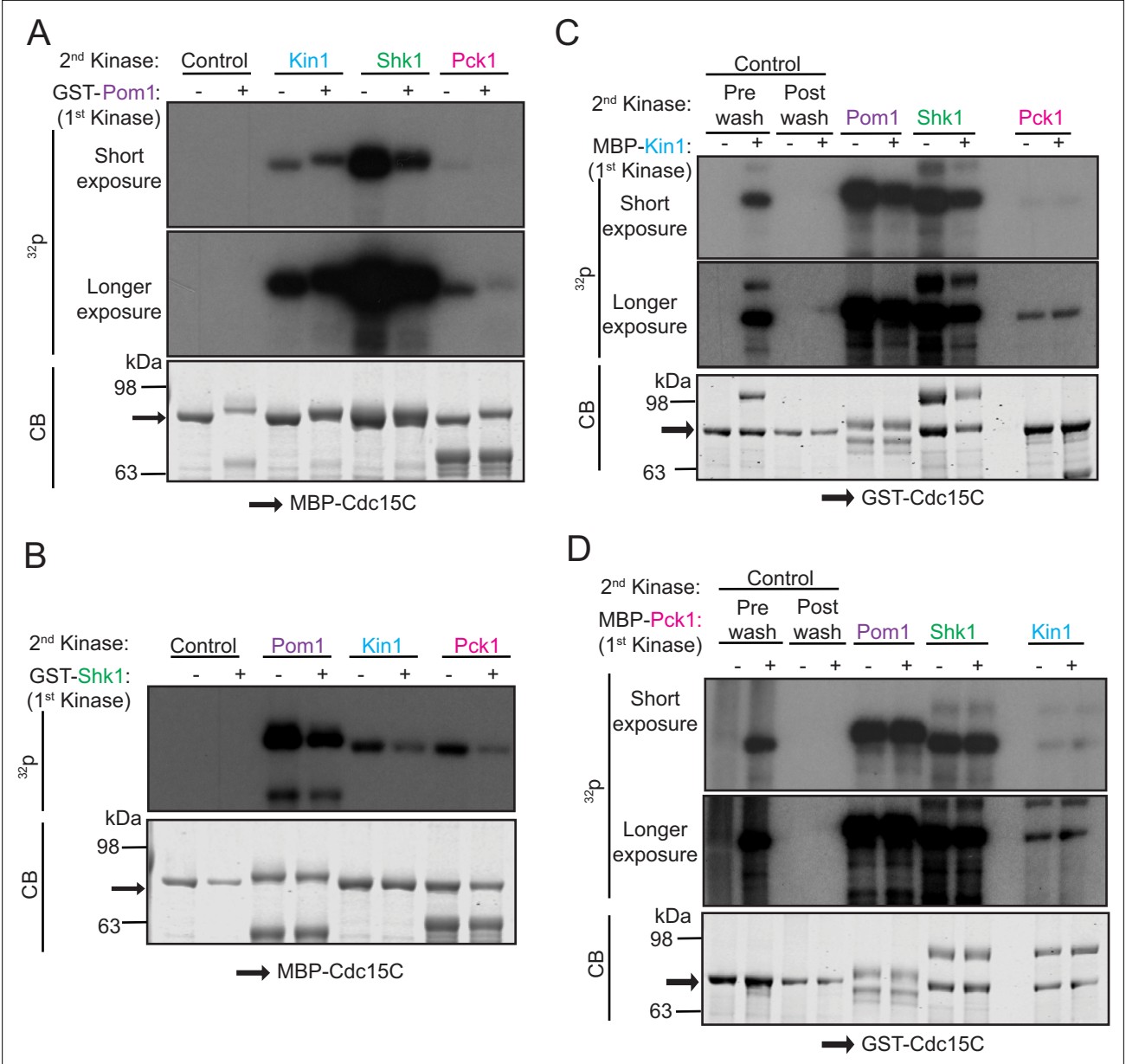

**Figure 2.** Polarity kinases phosphorylate overlapping sites on Cdc15. (A–D) In vitro kinase assays were performed using recombinant GST-Pom1 (A), GST-Shk1 (B), MBP-Kin1 (C), or MBP-Pck1 (D) with the indicated substrate (MBP-Cdc15C or GST-Cdc15C). After removing the first kinase, a second in vitro kinase assay was done with one of the other three kinases in the presence of radiolabeled γ-$^{32}$P-ATP. Kinase reactions were analyzed by SDS-PAGE, stained with CB (bottom) and $^{32}$P incorporation was detected by autoradiography (top and middle). The indicated molecular mass markers are in kDa. Arrows indicate MBP-Cdc15C or GST-Cdc15C.

The online version of this article includes the following source data and figure supplement(s) for figure 2:

**Source data 1.** Raw Coomassie stained gel images and full autoradiography images with the labeled band of interest.

**Figure supplement 1.** In vitro kinase assays show Cdc15C is directly phosphorylated on its C-terminus by Kin1, Shk1 or Pck1.

**Figure supplement 1—source data 1.** Raw Coomassie stained gel images and full autoradiography images with the labeled band of interest.

To test more deliberately whether any of these kinases influence Cdc15 phosphorylation by the others and whether they phosphorylate different sites, we carried out assays with each kinase in the presence of unlabeled ATP, removed the first kinase and then asked whether any of the other three kinases could phosphorylate pretreated Cdc15C (*Figure 2*). We found reduced phosphorylation of Cdc15C by Shk1 and Pck1 after pre-phosphorylation by Pom1 (*Figure 2A*). When Shk1 was used

as the first kinase, there was reduced Cdc15 phosphorylation by Pom1, Kin1, and Pck1 (*Figure 2B*). Cdc15C pre-phosphorylated by either Kin1 or Pck1 showed reduced phosphorylation by Pom1 or Shk1 (*Figure 2C and D*). However, when Cdc15C was phosphorylated by Kin1 first, it did not affect subsequent phosphorylation by Pck1, or vice versa (*Figure 2C and D*). Taken together, our findings indicate that there is a complex pattern of Cdc15 phosphorylation by these four kinases with both shared and unique sites.

To define the exact phosphorylation sites that can be targeted by each kinase, we used a combination of mass spectrometry and tryptic phosphopeptide mapping. In contrast to Pom1, which phosphorylates 22 sites on Cdc15 in vitro (*Bhattacharjee et al., 2020*), Kin1 and Shk1 phosphorylated 5 and 11 sites, respectively, all within the IDR (*Figure 3A*, *Figure 3—figure supplements 1 and 2*). Pck1 appeared to have a single major site of phosphorylation (S668) (*Figure 3A*, *Figure 3—figure supplement 3*). Validating the assignment of these sites, Kin1, Shk1, and Pck1 were unable to phosphorylate Cdc15C in which the relevant identified phosphorylation sites had been substituted with alanines (*Figure 3B–D*). A comparison of the sites phosphorylated by each of the four polarity kinases in vitro led us to conclude that multiple sites are shared among the kinases, although three kinases also phosphorylate unique sites consistent with previous data (*Figure 3*; *Kettenbach et al., 2015*; *Lee et al., 2018*; *Magliozzi et al., 2020*).

## Cdc15 mutants that prevent phosphorylation at sites targeted by polarity kinases localize more to the CR and alter cytokinesis timing

To evaluate the role of Cdc15 phosphorylation by the polarity kinases, we mutated the identified residues to alanine (to abolish phosphorylation) and integrated the series of phosphomutant alleles at the endogenous *cdc15* locus with or without an N-terminal mNeonGreen (mNG) tag. We chose not to generate asparate mutants because we previously found that they neither recapitulate the phosphorylated protein in charge nor retain the ability to bind 14-3-3 proteins (*Roberts-Galbraith and Gould, 2010*), as is typical for S/T to D mutants (*Dephoure et al., 2013*). Consistent with results from inhibiting all polarity kinases (*Figure 1*), replacing the complement of 31 phosphorylation sites in Cdc15 identified above with alanine led to a significant increase in Cdc15's electrophoretic mobility (*Figure 3E*). While phosphatase treatment of the Cdc15-31A mutant revealed that it is still phosphorylated (*Figure 3E*), the majority of wildtype Cdc15 is never so dephosphorylated, even during mitosis and cytokinesis (*Figure 3F*).

We next examined the effect of preventing phosphorylation on Cdc15 localization during cytokinesis. We found that there was more Cdc15-31A compared to Cdc15 in both the fully formed pre-constriction CR and in partially constricted CRs (*Figure 4A*). We also found more Rlc1, the regulatory light chain of myosin II, in CRs of *cdc15-31A* cells relative to wildtype cells (*Figure 4B*), indicating that the additional Cdc15-31A recruited more myosin II into CRs. Time-lapse imaging with Rlc1-mNG and Sid4-mNG as markers of the CR and spindle pole body (SPB), respectively, showed that *cdc15-31A* cells had altered cytokinesis dynamics. Although the length of CR formation (SPB separation to detection of a complete ring) was similar in wildtype and *cdc15-31A*, the periods of maturation (interval between CR formation and constriction initiation) and constriction (start to end of CR diameter decrease) were shorter in *cdc15-31A* (*Figure 4C and D*), in accord with the increased level of myosin II in the CR (*Calvert et al., 2011*).

## Kin1, Shk1, and Pck1 kinases inhibit Cdc15 membrane localization

Our finding of increased Cdc15-31A in the CR was consistent with previous findings that Pom1-mediated Cdc15 phosphorylation inhibits its PM localization as part of a mechanism Pom1 uses to prevent CR formation and septation at cell tips (*Bhattacharjee et al., 2020*). Therefore, we reasoned that if Kin1, Shk1, or Pck1 were inhibited during interphase, Cdc15, which is primarily hyperphosphorylated and cytosolic at this stage, would change its cellular localization. To test this, we examined the spatial distribution of mNG-Cdc15 in each of the analog-sensitive mutants of the four kinases individually, or in combination, in interphase cells that do not contain CRs. The ratio of mNG-Cdc15 PM to cytoplasmic localization increased when any of the kinases was inhibited and was highest in the strain in which Pom1, Shk1, and Kin1 activities were all inhibited (*Figure 5A*, *Figure 5—figure supplement 1A*). Like Pom1, Shk1 and Kin1 also play roles in preventing tip septation when the positive cue for medial septation, Mid1, is missing (*Cadou et al., 2010*; *Magliozzi et al., 2020*), and

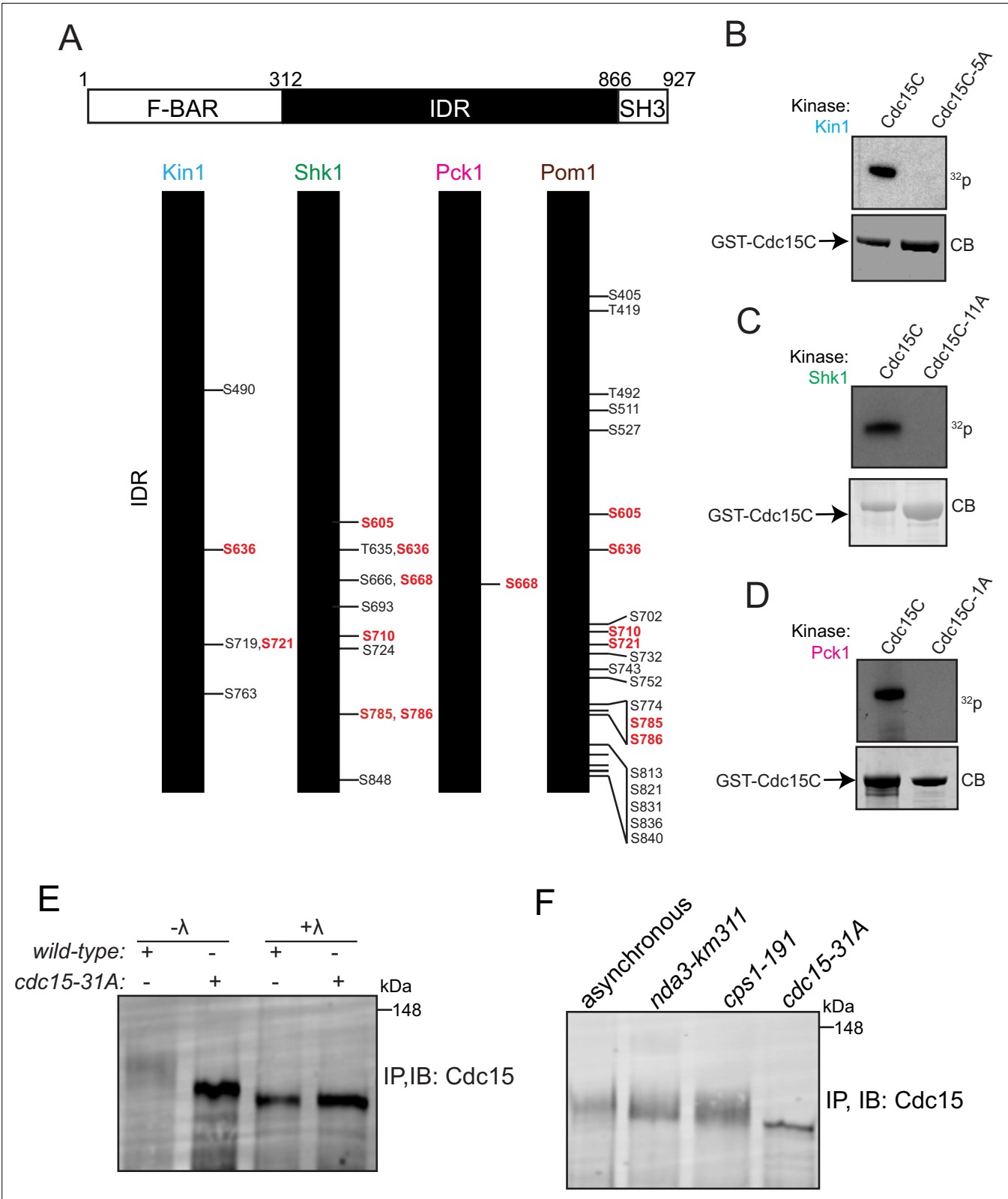

**Figure 3.** Identification of Cdc15 phosphosites for Kin1, Shk1, and Pck1. (**A**) Schematic of Cdc15 phosphorylation mutants. Numbers in red type indicate phosphorylation sites common to two or more kinases. (**B–D**) In vitro kinase assays using recombinant GST-Kin1 (**B**), GST-Shk1 (**C**), or MBP-Pck1 (**D**), and wildtype or mutant GST-Cdc15C proteins as substrates. All kinase assays were done in the presence of radiolabeled γ-$^{32}$P-ATP and the reactions were separated by SDS-PAGE, stained with CB (bottom) and $^{32}$P incorporation was detected by autoradiography (top). (**E**) Immunoprecipitation (IP) of Cdc15

*Figure 3 continued on next page*

*Figure 3 continued*

of the indicated strain, which was then treated with λ - phosphatase ( λ ) or buffer control. (**F**) Cell lysates prepared from the indicated genotypes and arrests were subjected to IP for Cdc15. For (**E, F**), samples were separated by SDS-PAGE and immunoblotted for Cdc15. The indicated molecular mass markers are in kDa.

The online version of this article includes the following source data and figure supplement(s) for figure 3:

**Source data 1.** Raw Western blot images, Coomassie stained gel images and full autoradiography images with the labeled band of interest.

**Figure supplement 1.** Mapping of phosphorylation sites of Kin1, Shk1, and Pck1 on Cdc15.

**Figure supplement 1—source data 1.** Full autoradiography images from mapping of phosphorylation sites of Kin1, Shk1, and Pck1 on Cdc15.

**Figure supplement 2.** Mapping of phosphorylation sites of Kin1, Shk1, and Pck1 on Cdc15.

**Figure supplement 2—source data 1.** This source file contains autoradiographs of phosphopeptide maps.

**Figure supplement 3.** Mapping of phosphorylation sites of Kin1, Shk1, and Pck1 on Cdc15.

**Figure supplement 3—source data 1.** This source file contains autoradiographs of phosphopeptide maps.

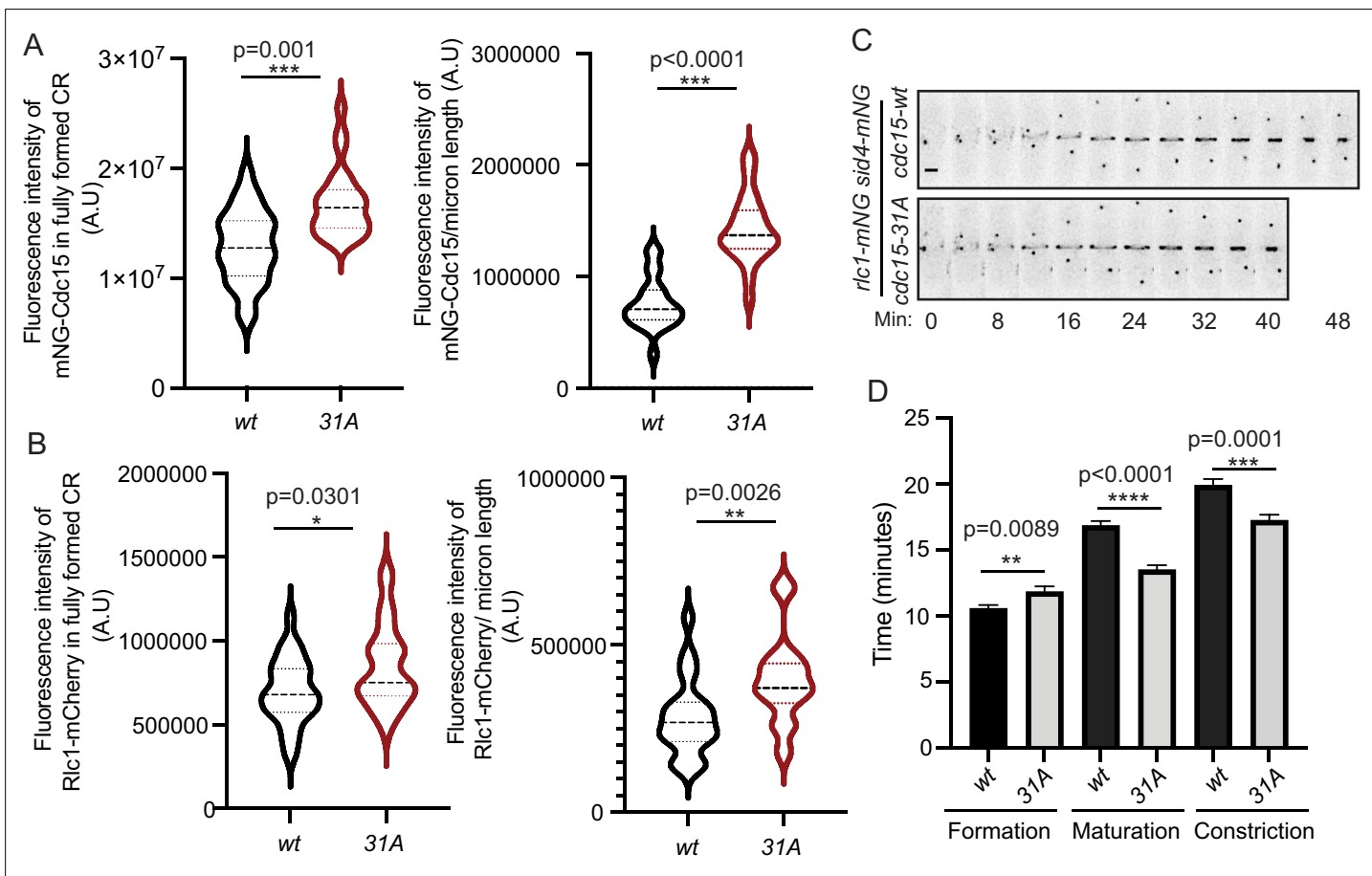

**Figure 4.** Quantification of mNG-Cdc15 abundance in the cytokinetic ring (CR) and cytokinesis dynamics of Cdc15 phosphomutants. (**A, B**) Quantification of the fluorescence intensities of Cdc15 (**A**) or Rlc1 (**B**) in the fully formed CR (left) and constricting CRs (right) of wildtype and *cdc15-31A* cells. The fluorescence intensities of the CRs were normalized to the respective whole cell intensities. (**A**) n ≥ 29 cells and (**B**) n ≥ 26 cells (N = 3). (**C**) Representative montages from live-cell time-lapse movies of the indicated strains. Images were acquired every 2 min and every 4 min are shown. Numbers indicate min from SPB separation. Scale bar, 2 µm. (**D**) Quantification of the length of cytokinesis of the indicated strains (N = 3), n ≥ 21 cells. Error bars indicate SEM. All statistical comparisons were made using non-parametric *t*-test with Mann–Whitney test. *p<0.05, **p<0.01, ***p<0.001and ****p<0.0001.

The online version of this article includes the following source data for figure 4:

**Source data 1.** Raw xlsx and pzfx files used for analysis of the dataset.

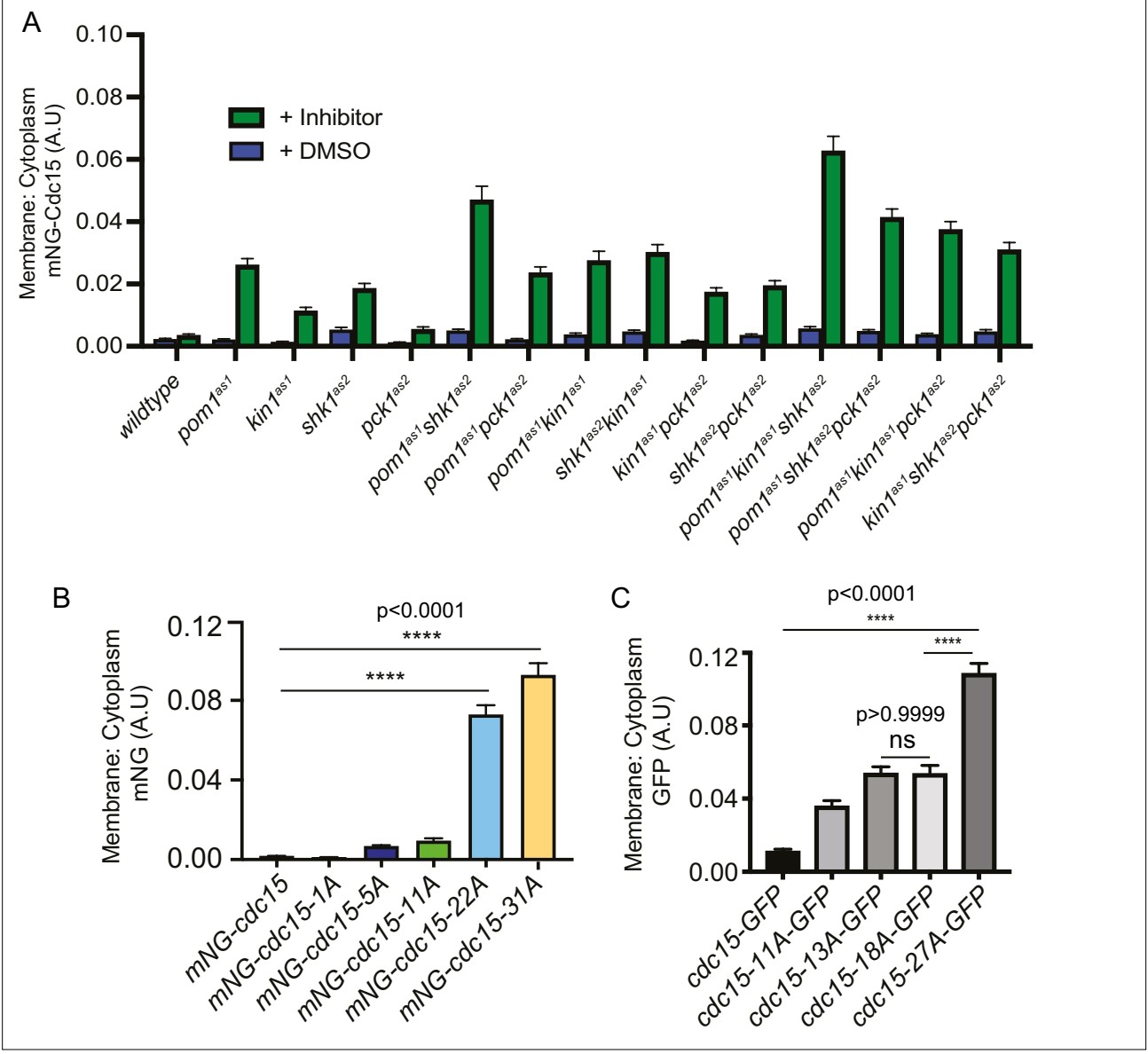

**Figure 5.** Ratio of membrane to cytoplasm localization of Cdc15 and Cdc15 phosphomutants. (**A**) Ratio of membrane to cytoplasm location of mNG-Cdc15 in arbitrary units (A.U.) in the indicated strains after treatment with inhibitor(s) 15 µM 3-MB-PP1 and/or 30 µM 3BrB-PP1 for 15 min. n ≥ 97 cells. (**B,** **C**) Ratio of membrane to cytoplasm location of mNG-Cdc15 in the indicated strains. n ≥ 65 (**B**) and n ≥ 97 cells (**C**), respectively. Error bars indicate SEM (N = 3). Comparisons were made using one-way ANOVA with Tukey's multiple-comparison test. ****p<0.0001 and ns = nonsignificant, p>0.05.

The online version of this article includes the following source data and figure supplement(s) for figure 5:

**Source data 1.** Raw xlsx and pzfx files used for analysis of the dataset.

**Figure supplement 1.** mNG-Cdc15 localization after treatment with DMSO or inhibitor(s) and in Cdc15 phosphomutants.

**Figure supplement 2.** Quantification of tip-septa of the indicated strains.

**Figure supplement 2—source data 1.** Raw xlsx and pzfx files used for analysis of the dataset.

we found that inhibition of Pck1 also allowed a small increase in tip septation in a *mid1Δ* background (*Figure 5—figure supplement 2A*). The percentage of tip septa formed when either Kin1^as1, Shk1^as, or Pck1^as2 were inhibited in *mid1Δ* cells correlated with the extent of increased membrane localization (*Figure 5—figure supplement 2A*).

Because the ratio of PM to cytoplasmic localization of interphase cells appeared to increase as a function of the number of Cdc15 sites phosphorylated by each kinase (*Figure 5A*), we next examined

the localization of the mNG-Cdc15 phosphomutants. Paralleling the results of kinase inhibition, we found that the percentage of cortical Cdc15 localization increased with the number of phospho-sites mutated to alanine, with a significant jump between the 11A and the 22A mutants (*Figure 5B*, *Figure 5—figure supplement 1B*). These results are also concordant with finding more Cdc15-31A in the CR (*Figure 4A*).

The significant increase in cortical localization between 11A and 22A mutations (*Figure 5B*) prompted us to investigate the intracellular localization of a different set of Cdc15 phosphomutants during interphase (*Roberts-Galbraith and Gould, 2010*). These initial mutants were tagged with GFP at their C-termini. While the results cannot be compared directly to the mNG, N-terminally tagged mutants, we again observed a significant increase in cortical localization in the 27A mutant compared to the 11A, 13A, or 18A mutants (*Figure 5C*). Inspection of the phosphorylation sites mutated in each of the mutants did not indicate that loss of specific sites or lack of phosphorylation of any specific IDR region correlated with the greatest localization change. Instead, the analysis suggested that a threshold of dephosphorylation may trigger a change in protein conformation and ability to bind membrane. We used modeling to explore this hypothesis.

## Simulation of phospho-regulated Cdc15 conformational changes

To better understand how IDR phosphorylation regulates Cdc15, we performed coarse-grained molecular dynamics simulations in which each amino acid is represented by a single bead on the α carbon location while the solvent is represented implicitly. Interactions between beads were given by the Hydrophobicity Scale (HPS) model, developed to match the radius of gyration ($R_G$) for multiple IDRs (*Dignon et al., 2018*), based on the Kapcha–Rossky hydropathy scale (*Kapcha and Rossky, 2014*). We further used an extension of the HPS model that includes the effects of phosphorylation and associated change in charge interactions (*Perdikari et al., 2021*). In our Cdc15 dimer model, the folded F-BAR and SH3 domains were fixed rigidly according to the recently determined crystal structure of the Cdc15 F-BAR domain (PDB 6XJ1; *Snider et al., 2020*) and the AlphaFold model of the Cdc15 SH3 domain (AF-Q09822-F1; *Jumper et al., 2021*; *Varadi et al., 2022*). All other amino acids were connected by flexible linkers with equilibrium length 3.81 Å (*Figure 6A*). Assuming HPS interactions between all beads in the system, we simulated Cdc15 dimers in different phosphorylation states: dephosphorylated, phosphorylated at all 31 sites on each IDR in the dimer, and at the sites for the individual kinases Kin1 (5 sites), Shk1 (11 sites), and Pom1 (22 sites). To check that the system approaches the same equilibrium, independent of the starting configuration, we performed simulations where IDRs were initially separated or where IDRs were initially interacting with each other (*Figure 6A*, *Figure 6—figure supplement 1A and B*).

Simulations showed that increasing phosphorylation drives a large structural transition in the Cdc15 dimer. In the dephosphorylated state, the IDRs preferred to interact with each other but progressive phosphorylation weakens this preference and drives the IDRs to the poles of the F-BAR (*Figure 6B*, *Figure 6—video 1* and *Figure 6—video 2*). Furthermore, simulations suggest the IDR makes contacts with F-BAR tips (*Figure 6—figure supplements 2 and 3A*). Although separation of the IDRs with increasing phosphorylation is gradual, visual inspection shows it begins to occur when the distance between the IDR center of mass (COM) of each chain exceeds 100 Å (*Figure 6—figure supplement 1C*). Furthermore, we see a significant increase in the fraction of time the IDRs spend with high COM separation for the Pom1 and fully phosphorylated cases (*Figure 6C and D*). Additionally, we observe that the SH3 domains make contacts with the F-BAR domain in similar regions to the IDR, and these contacts are also regulated by phosphorylation (*Figure 6—figure supplements 2 and 3B*). These simulation results suggest that phosphorylation inhibits Cdc15 by driving the IDR and SH3 regions towards the tips of the F-BAR domain, potentially blocking its end-to-end oligomerization and weakening PM avidity.

We repeated the Cdc15 dimer simulations using a different α carbon coarse-grained model previously applied to IDRs (*Dignon et al., 2018*): the Kim and Hummer model D (KH D) (*Kim et al., 2008*) based on the Miyazawa–Jernigan potential (*Miyazawa and Jernigan, 1996*; see 'Materials and methods'). The trends revealed by the HPS model were confirmed; however, the transitions between low and high IDR COM separation were more abrupt (*Figure 6—figure supplement 1D and F*). Unlike the HPS case, the systems did not fully equilibrate over comparable timescales, retaining memory of their separated or interacting starting IDR configurations (see 'Discussion').

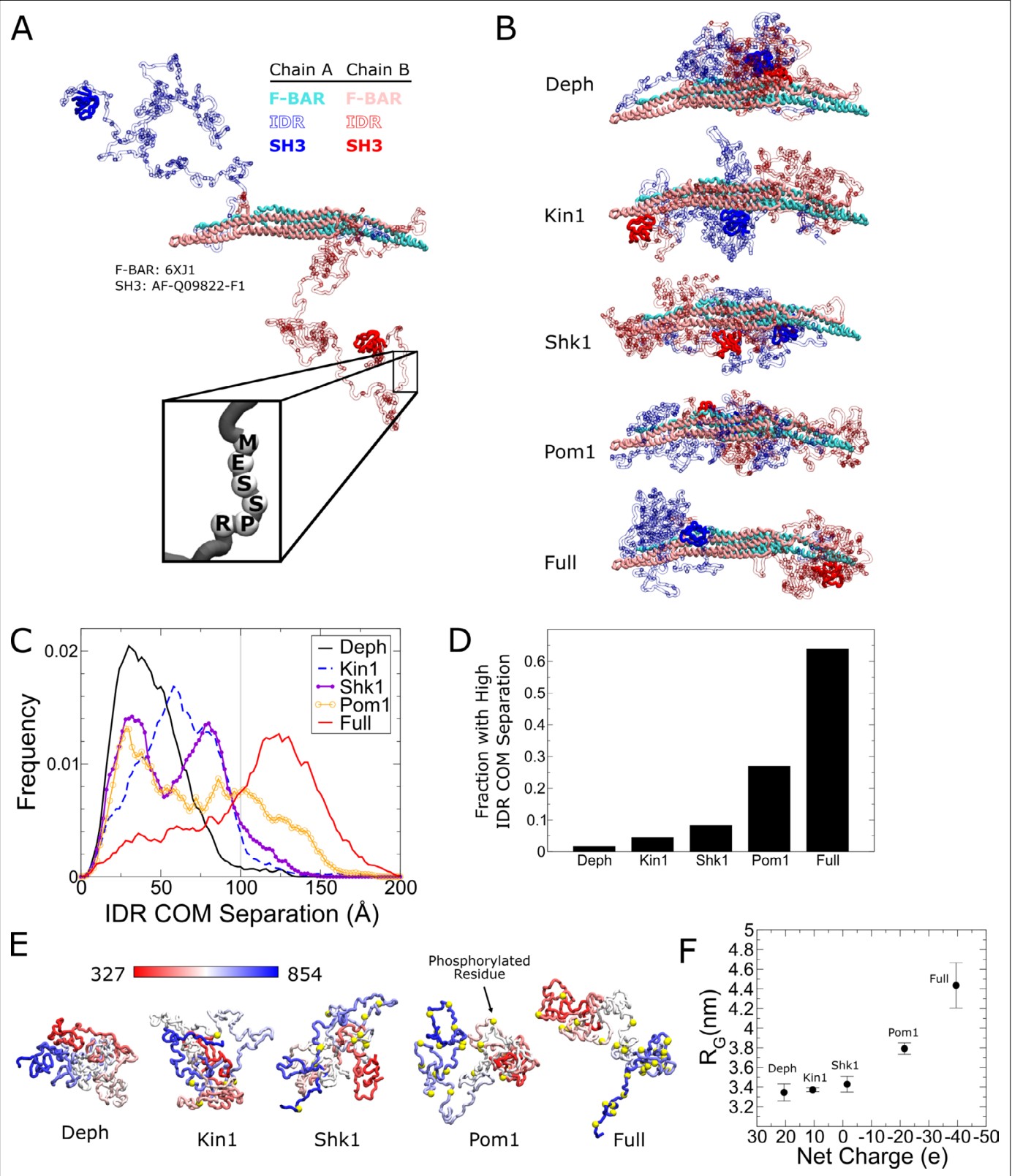

**Figure 6.** Coarse-grained molecular dynamics of Cdc15 using the Hydrophobicity Scale (HPS) model show large conformational transition induced by Cdc15 intrinsically disordered region (IDR) phosphorylation. (**A**) Snapshot of an α carbon representation of a Cdc15 dimer with each amino acid represented by a single bead. The F-BAR and SH3 regions are kept rigid (PDB: 6XJ1 and AlphaFold model AF-Q09822-F1, respectively). All other residues connected with flexible linkers (inset). Snapshot shows the initial condition with the IDRs started apart from each other. (**B**) Representative

*Figure 6 continued on next page*

*Figure 6 continued*

simulation snapshots of the indicated phosphorylation state: Deph has no phosphorylation, Full includes all sites from *Figure 3*, otherwise the label states that the model includes all sites of the named kinase. (C) Frequency distribution of distance between the center of mass (COM) of each IDR. Gray line indicates 100 Å. Ten independent simulations of at least 1.9 μs were performed for each phosphorylation state with half initialized with the IDRs apart (panel **A** or *Figure 6—figure supplement 5A*) and half with the IDRs interacting (*Figure 6—figure supplement 5B*). (D) Fraction of trajectories of indicated states with COM separation greater than 100 Å. (E) Representative snapshots of simulations of single Cdc15 IDR (residues 327–854, connected with flexible linkers) in different phosphorylation states. (F) Average radius of gyration, $R_G$, versus net charge of the chain (phosphorylation charge treated as –2e.) Error bars as described in 'Materials and methods'.

The online version of this article includes the following video, source data, and figure supplement(s) for figure 6:

**Source data 1.** Raw csv files used to plot the dataset.

**Figure supplement 1.** Comparison of Hydrophobicity Scale (HPS) and Kim and Hummer model D (KH D) for Cdc15 dimers and convergence properties of simulations.

**Figure supplement 1—source data 1.** Raw csv files used to plot the dataset.

**Figure supplement 2.** Cdc15 dimer simulation contact maps in the Hydrophobicity Scale (HPS) model.

**Figure supplement 2—source data 1.** Raw data files used to plot the dataset.

**Figure supplement 3.** Cdc15 dimer simulation F-BAR contacts in the Hydrophobicity Scale (HPS) model.

**Figure supplement 3—source data 1.** Raw xyz files used to plot the dataset.

**Figure supplement 4.** Cdc15 intrinsically disordered region (IDR) size in single chain simulations.

**Figure supplement 4—source data 1.** Raw csv files used to plot the dataset.

**Figure supplement 5.** Diagram of states for Cdc15 intrinsically disordered region (IDR).

**Figure supplement 5—source data 1.** Raw csv files used to plot the dataset.

**Figure 6—video 1.** Simulation of Cdc15 dimer in the Hydrophobicity Scale (HPS) model with intrinsically disordered regions (IDRs) started apart.
https://elifesciences.org/articles/83062/figures#fig6video1

**Figure 6—video 2.** Simulation of Cdc15 dimer in the Hydrophobicity Scale (HPS) model with intrinsically disordered regions (IDRs) started interacting.
https://elifesciences.org/articles/83062/figures#fig6video2

To investigate how phosphorylation regulates the expansion of the Cdc15 IDR, we performed coarse-grained molecular dynamics simulations of the Cdc15 IDR (residues 327–854) in isolation. For the HPS model, dephosphorylated Cdc15 IDR is relatively collapsed and undergoes progressive expansion with increased phosphorylation, which is more pronounced for the Pom1 and fully phosphorylated cases (*Figure 6E and F*). In simulations that show the coil to globule transition with increasing temperature, we found that the phosphorylation-induced relative change in $R_G$ is maximal near room temperature (*Figure 6—figure supplement 4A and B*). Similar results were obtained with the KH D model, although the change in IDR size was smaller and the coil-to-globule transition happened at higher temperatures compared to HPS (*Figure 6—figure supplement 4C and D*). In order to determine the effect of the phosphorylation charge on these results, we simulated the fully phosphorylated case with –1.5e charge per phosphorylation, instead of –2e, for both models (the average value can vary between –1e and –2e at neutral pH based on the residues measured pKa values; *Bienkiewicz and Lumb, 1999*). The expansion is commensurate with net charge added by phosphorylation (*Figure 6F*, *Figure 6—figure supplement 5*), in agreement with the diagram of states of *Das and Pappu, 2013* that predicts an increase in negative charge due to phosphorylation would cause the Cdc15 IDR to transition from a collapsed globule or tadpole closer to a random coil, hairpin or chimera (*Figure 6—figure supplement 5*).

## Cdc15 IDR undergoes phosphoregulated phase separation

The collapse of the dephosphorylated IDRs into a medial globule in our dimer simulations and the relatively compact size of the single dephosphorylated IDR simulations suggest the possibility that the dephosphorylated Cdc15 IDR regions undergo phase separation with other Cdc15 dimers and/or other CR proteins. Indeed, a reduction in $R_G$, is a good indicator of intrinsically disordered protein tendency to phase-separate (*Lin and Chan, 2017*). Increasing phosphorylation of the Cdc15 IDR phosphorylation might disrupt phase separation, similar to multisite phosphorylation of the low sequence-complexity region of FUS (*Monahan et al., 2017*).

To experimentally test whether Cdc15 IDR can undergo LLPS regulated by IDR phosphorylation, we first expressed recombinant Cdc15-IDR-SH3 (residues 327–927) together with Pom1 to ensure that the IDR became highly phosphorylated (*Bhattacharjee et al., 2020*). In these first experiments, the F-BAR domain was omitted because of its propensity to oligomerize at physiological salt concentrations in vitro (*McDonald et al., 2015*; *Snider et al., 2020*). Pom1-phosphorylated Cdc15 IDR-SH3 was purified at high salt concentration (*Figure 7—figure supplement 1A*), labeled with Alexa Fluor-488, and treated with $\lambda$-phosphatase or buffer control (*Figure 7—figure supplement 1B*). The salt concentration was reduced to 150 mM NaCl and a molecular crowding agent (5% PEG, see 'Materials and methods') was added in certain experiments, as indicated. Droplet formation was observed and measured by fluorescence microscopy. We found that Cdc15-IDR-SH3 formed droplets in a manner that depended on protein and salt concentrations (*Figure 7A and B*). Importantly, phase separation was observed only when Cdc15-IDR-SH3 was dephosphorylated (*Figure 7A and B*). Using the optimal protein concentration derived from phase diagrams (25 µM) (*Figure 7A*), we observed that Cdc15-IDR-SH3 droplets could fuse, consistent with a dynamic liquid-like nature (*Figure 7C* and *Figure 7—video 1*). Fluorescence recovery after photobleaching (FRAP) analysis of Cdc15-IDR-SH3 droplets in the absence of PEG showed that the droplets were dynamic (*Figure 7D*). Furthermore, line-scans through the center of each droplet confirmed that dephosphorylated Cdc15-IDR-SH3 incorporated homogenously into the droplets (*Figure 7E and F*).

We next tested whether Cdc15-IDR-SH3 would phase-separate if attached to supported lipid bilayers. For this, a His tag was appended to Cdc15-IDR-SH3 and Ni-NTA-tagged lipids were included in the bilayers. Similar to the in-solution results, dephosphorylated membrane-bound His-Cdc15-IDR-SH3 consistently formed condensates (*Figure 7G*) in a protein and salt concentration-dependent manner (*Figure 7H*).

Via its SH3 domain, Cdc15 binds Fic1 regardless of Cdc15 phosphostate and also paxillin-like Pxl1 when Cdc15 is dephosphorylated (*Roberts-Galbraith et al., 2009*; *Bhattacharjee et al., 2020*). The single PxxP motif in Pxl1 that binds the Cdc15 SH3 domain has been mapped and the interaction with Cdc15 SH3 recapitulated using a synthetic peptide consisting of Pxl1 residues 177–188 (*Snider et al., 2022*). We sought to determine whether phase-separated droplets of dephosphorylated Cdc15-IDR-SH3 could recruit Pxl1 peptide and/or recombinant Fic1 (*Figure 8—figure supplement 1A*). When Pom1-phosphorylated Cdc15-IDR-SH3 was combined with equal concentrations of either Fic1 or the Pxl1 peptide, we observed droplets containing both molecules but only when Cdc15-IDR-SH3 was dephosphorylated (*Figure 8A and B*). When all three proteins were combined, we detected droplets containing all three (*Figure 8C*). To make sure that incorporation of other proteins into Cdc15-IDR-SH3 droplets was specific, we tested whether recombinant mCherry (*Figure 8—figure supplement 1B*) would incorporate into the droplets. It did not (*Figure 8D*). As an additional control, we produced Pom1-phosphorylated Cdc15-IDR-SH3 with a mutation (W903S) that prevents Fic1 binding (*Bhattacharjee et al., 2020*; *Figure 8—figure supplement 1B*) and asked whether Fic1 would incorporate into droplets with it upon phosphatase treatment. It did not (*Figure 8E*).

We next asked whether full-length Cdc15 would also undergo LLPS in solution when dephosphorylated. For this, we used an F-BAR oligomerization mutant (E30K, E152K) (*Figure 9—figure supplement 1A*) to prevent extensive F-BAR-mediated oligomerization in vitro. In these assays, in addition to the Pxl1 peptide and Fic1, we also added a Cdc12 formin peptide (residues 20–40) that directly binds the Cdc15 F-BAR domain with high affinity (*Willet et al., 2015a*). When we combined labeled dephosphorylated Cdc15-E30K,E152K with labeled Fic1 and the Pxl1 and Cdc12 peptides (one or the other peptide in the mix labeled) in equimolar ratio, we observed droplets containing multiple components that were not observed if Cdc15-E30K, E152K remained phosphorylated (*Figure 9A and B*). With only three of the four components labeled, we observed that 84% of droplets contained at least three components (Cdc15, Fic1, and Pxl1 or Cdc15, Fic1, and Cdc12) (*Figure 9A and B*). The remainder of droplets contained either labeled Cdc15 and Fic1, Cdc15 and Pxl1, or Cdc15 and Cdc12. Therefore, we conclude that dephosphorylated full-length Cdc15 can recruit its binding partners into phase-separated droplets.

## Cdc15 condensates in cells exhibit liquid-like properties

Our observations of dephosphorylated Cdc15 condensate formation in vitro was reminiscent of previous results showing that several different Cdc15 phosphoablating mutants localized in discrete

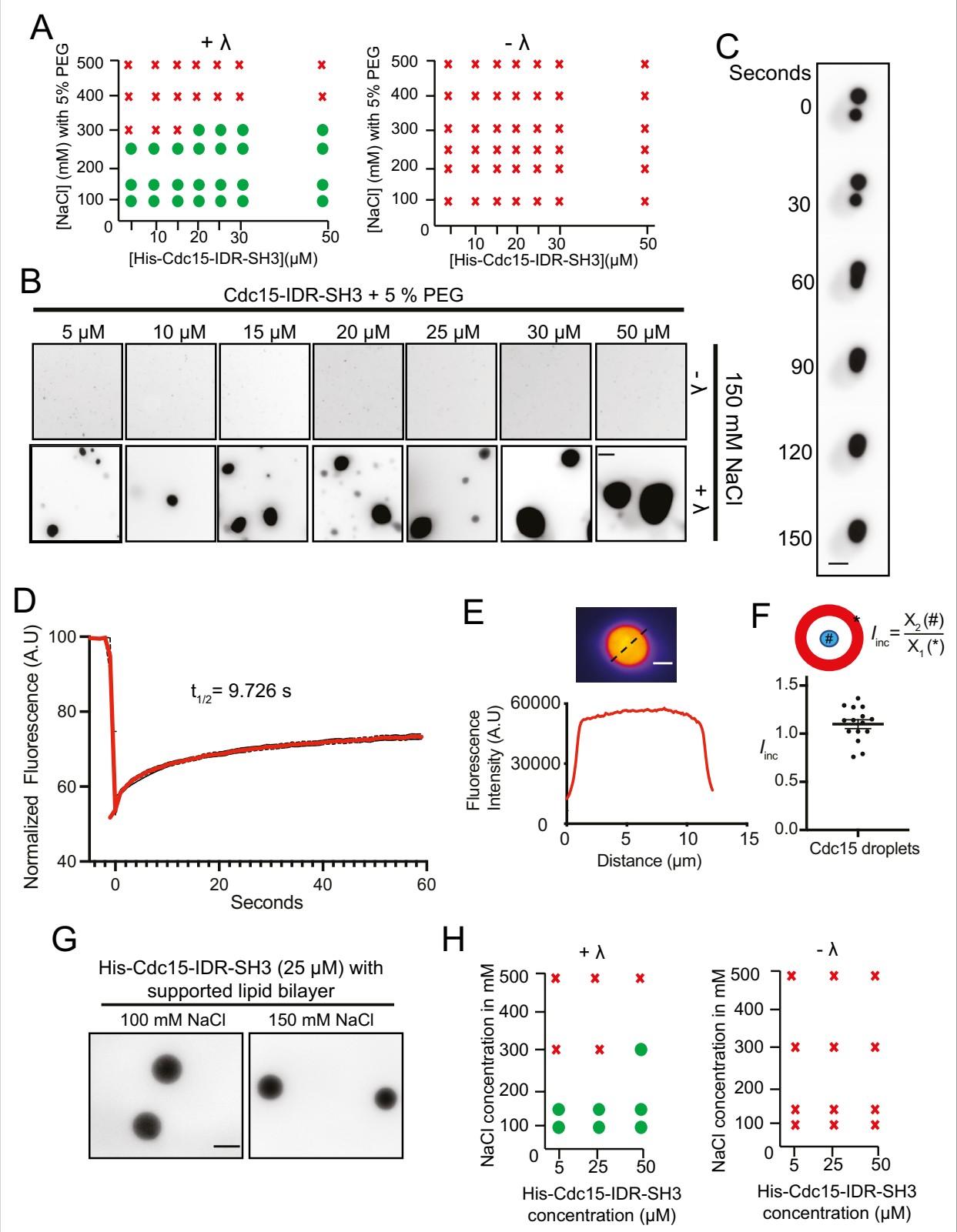

**Figure 7.** Cdc15-IDR-SH3 phase separates in a phosphoregulated manner. (**A**) Phase diagrams showing whether purified Pom1-phosphorylated Cdc15-IDR-SH3 labeled with Alexa-488 succinimide ester dye formed droplets at different protein concentrations, NaCl concentrations, and with or without λ-phosphatase (λ) treatment. (**B**) In vitro droplet formation at different concentrations of Pom1-phosphorylated Cdc15-IDR-SH3 in 50 mM Tris pH 7.4, 150 mM NaCl with 5% PEG after treatment with λ (lower panel). Scale bar, 10 μm. (**C**) Time-lapse analysis showing fusion of droplets containing Cdc15-

*Figure 7 continued on next page*

*Figure 7 continued*

IDR-SH3 after treatment with $\lambda$. Phase separation assays were performed in 50 mM Tris pH 7.4, 150 mM NaCl. Scale bar, 5 μm. (**D**) FRAP dynamics of Cdc15–IDR-SH3 condensates after partial bleaching. In these experiments, droplets were formed in 50 mM Tris pH 7.4, 150 mM NaCl without PEG and with 25 μM of labeled protein. n = 10, N = 3. (**E**) Fluorescent image (top) and line scan (bottom) performed through the center of a droplet. (**F**) Incorporation index ($I_{inc}$ = $Intensity_{center}$ / $Intensity_{edge}$) for Cdc15-IDR- Alexa-488 when dephosphorylated at 50 mM Tris pH 7.4, 150 mM NaCl, 5% PEG with 25 μM of protein. n = 15 droplets. (**G, H**) TIRF imaging was used to detect condensates of labeled His-Cdc15-IDR-SH3 on supported lipid bilayers that include Ni-NTA lipids. (**G**) Condensate formation at 25 μM His-Cdc15-IDR-SH3 on a supported lipid bilayer after treatment with $\lambda$ -phosphatase in 50 mM Tris pH 7.4, 250 mM NaCl. Scale bar, 5 μm. (**H**) Phase diagrams of purified His-Cdc15-IDR-SH3 labeled with Alexa-488 on a supported lipid bilayer.

The online version of this article includes the following video, source data, and figure supplement(s) for figure 7:

**Source data 1.** Mov file for *Figure 7C*, raw xlsx and pzfx files for *Figure 7D–F*.

**Figure supplement 1.** Cdc15-IDR-SH3 purification.

**Figure supplement 1—source data 1.** Raw Western blot images and Coomassie stained gel images with the labeled band of interest.

**Figure 7—video 1.** Fusion event of Cdc15-IDR-SH3 droplet represented at *Figure 7C*.

https://elifesciences.org/articles/83062/figures#fig7video1

puncta at the cortex of interphase cells (*Roberts-Galbraith et al., 2010*; *Bhattacharjee et al., 2020*), and wildtype Cdc15 in the CR has been observed to have a node-like organization by super-resolution imaging (*Laplante et al., 2016*; *McDonald et al., 2017*). Our new Cdc15 phosphomutants also formed cortical puncta or condensates during interphase, the number and size of which increased significantly in the 22A and 31A mutants (*Figure 10A and B*). We observed these condensates more closely and found that they were dynamic and underwent apparent fission and fusion events (*Figure 10C*, *Figure 9—video 1* and *Figure 9—video 2*), suggesting that they had some liquid-like properties. In *cdc15-31A* but not wildtype cells, Fic1 and Rlc1 were also recruited into these interphase condensates decorating the cortex (*Figure 10D*). Taken together, our data suggest that multisite phosphorylation of Cdc15 by polarity kinases inhibits Cdc15 condensate formation, Cdc15 membrane binding, and premature and non-medial CR construction.

## Discussion

The F-BAR protein Cdc15 is an essential CR component and a linker of the CR to the PM. Cdc15 is phosphoregulated in a cell cycle-specific manner, but how individual kinases contribute to this regulation and precisely how phosphorylation affects Cdc15 properties have not been clear. We presented evidence that progressive phosphorylation drives separation of the Cdc15 IDR regions, which antagonizes Cdc15 phase separation, condensate formation on the PM, and recruitment of other CR proteins. Leveraging a suite of Cdc15 protein kinases during interphase appears to ensure sufficient Cdc15 phosphorylation that CR assembly begins only at the proper place and time. More broadly, our work suggests phase separation as a mechanism involved in CR assembly.

The first detailed study of Cdc15 phosphorylation focused on phosphorylation sites that could be targeted by the Cdc14 family phosphatase, Clp1, and those matching a 14-3-3 binding site; this study estimated that ~35 sites within Cdc15 could be phosphorylated (*Roberts-Galbraith et al., 2010*). Subsequently, several kinases have been identified to affect Cdc15 phosphostatus, and 74 sites of Cdc15 phosphorylation have been reported from MS-based studies (*Chen et al., 2013*; *Kettenbach et al., 2015*; *Lee et al., 2018*; *Swaffer et al., 2018*; *Bhattacharjee et al., 2020*; *Magliozzi et al., 2020*; *Harris et al., 2022*). Based on gel shifts, the mutant generated in this study, Cdc15-31A, appears to be minimally phosphorylated compared to wild type protein. We reason therefore that the total number of Cdc15 phosphorylation sites is closer to 35 than 74 and that there are probably a number of mis-identified sites arising from the technical ambiguity of site assignment in peptides containing multiple potential phosphorylation sites. Indeed, there were numerous differences in the sites we identified for the polarity kinases compared to MS only-based studies. However, that Cdc15-31A retains some in vivo phosphorylation points to the existence of still another protein kinase(s) involved in Cdc15 phosphoregulation that contributes a unique set of targeted residues.

Given these technical challenges, it has been unclear exactly which Cdc15 sites each kinase phosphorylates and whether they affect the protein similarly. Our results indicate that there is little if any cooperativity or antagonism among the kinases and point to the total charge added or removed from

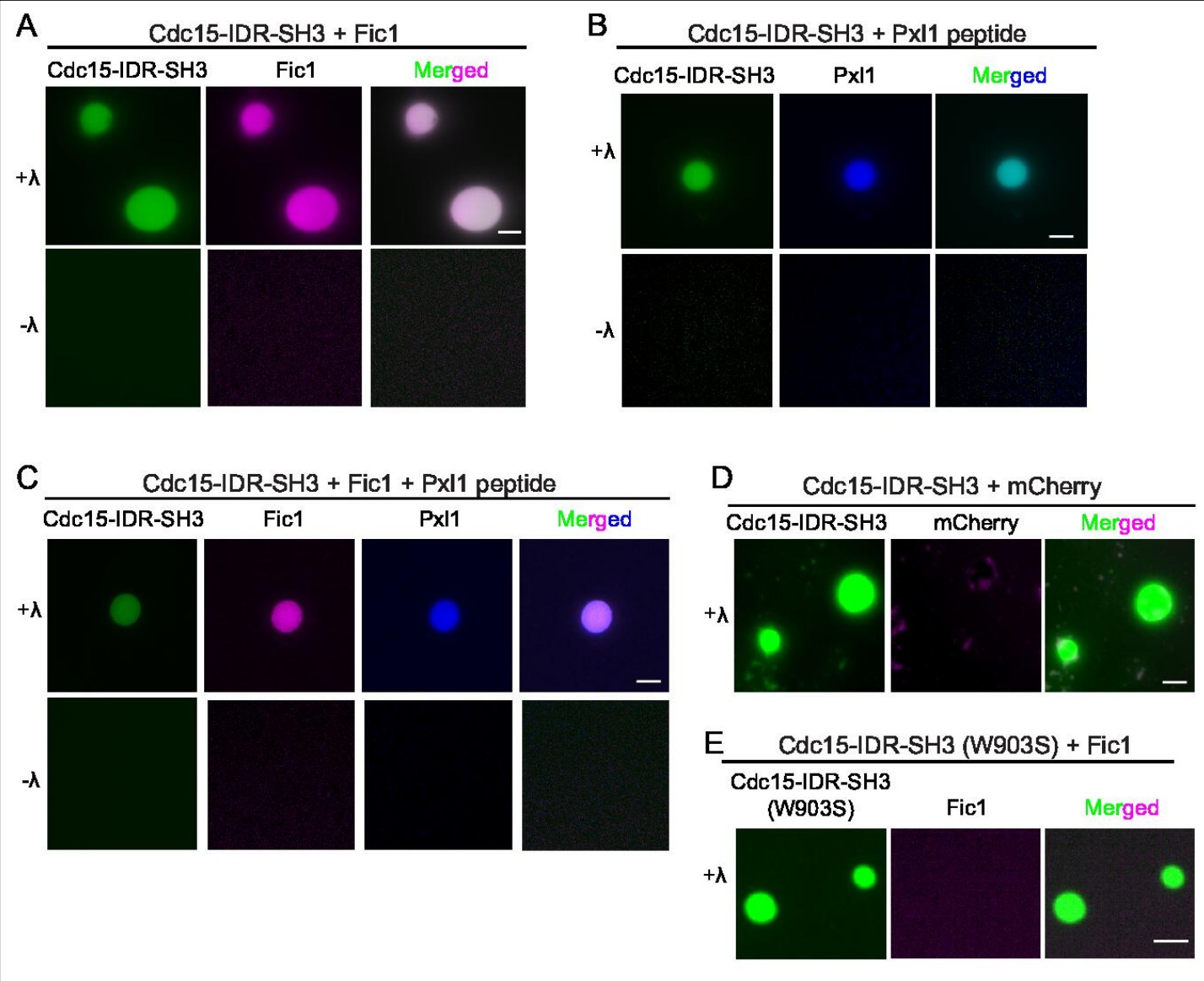

**Figure 8.** Cdc15 promotes in vitro phase separation of its SH3 domain binding partners, Fic1 and Pxl1. Images of droplets formed after (**A**) Pom1-phosphorylated Cdc15-IDR-SH3 (Alexa 488 labeled) and Fic1 (Alexa 647 labeled), (**B**) Pom1-phosphorylated Cdc15-IDR-SH3 (Alexa 488 labeled) and Pxl1 (residues 177–188; Alexa 405 labeled), or (**C**) Pom1-phosphorylated Cdc15-IDR-SH3 (Alexa 488 labeled), Fic1 (Alexa 647 labeled), and Pxl1 (residues 177–188; Alexa 405 labeled) were mixed at 10 µM concentration each and treated or not with $\lambda$-phosphatase ($\lambda$). No visible droplets were detected when Cdc15 remained phosphorylated. (**D**) Images of droplets formed when Pom1-phosphorylated Cdc15-IDR-SH3 (Alexa 488 labeled) and mCherry were mixed at 10 µM concentration each and then dephosphorylated with $\lambda$ phosphatase. (**E**) Images of droplets formed when Pom1-phosphorylated Cdc15-IDR-SH3 (W903S) (Alexa 488 labeled) and Fic1 (Alexa 647 labeled) were mixed at 10 µM concentration each and then treated with $\lambda$ phosphatase. (**A–E**) Assays were performed in 50 mM Tris pH 7.4, 150 mM NaCl, and 5% PEG. Scale bars, 5 µm.

The online version of this article includes the following source data and figure supplement(s) for figure 8:

**Figure supplement 1.** Purification of His-Fic1, His-mCherry, and His-Cdc15-IDR-SH3 (W903S).

**Figure supplement 1—source data 1.** Raw Coomassie stained gel images with the labeled band of interest.

the IDR as the most significant regulatory factor. The only specificity we found is the number of sites that can be phosphorylated by each kinase, assigned using a combination of technical approaches that is more accurate than an MS only-based approach. Our simulation results suggest that a certain threshold of phosphorylation must be reached to elicit a major change in IDR organization and IDR-IDR interaction, and we reason that there is an inherent limitation to how any single kinase can affect Cdc15 structure and function. This may explain why so many protein kinases are involved in Cdc15

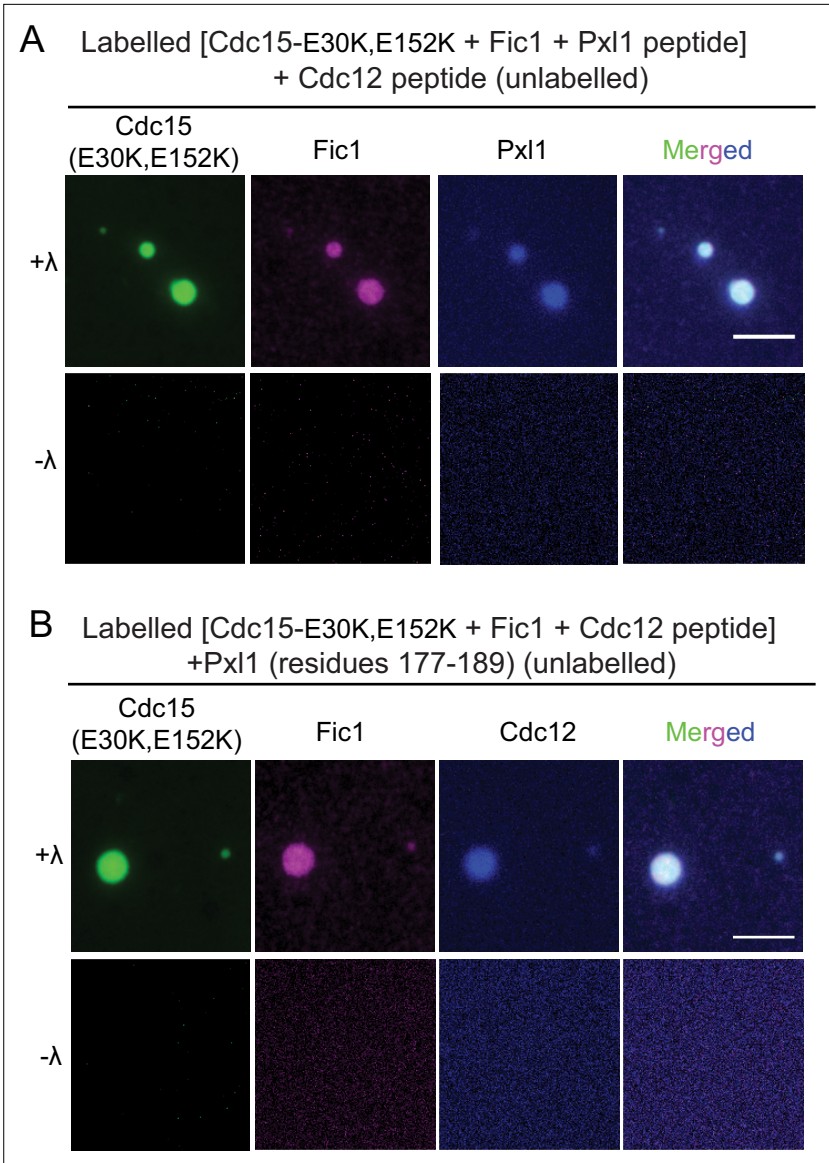

**Figure 9.** Full-length Cdc15 promotes in vitro phase separation of its F-BAR and SH3 domain binding partners. Images of droplets formed with (**A**) 10 µM each full-length Pom1-phosphorylated Cdc15 (E30K, E152K) (Alexa 488 labeled) treated or not with $\lambda$-phosphatase and incubated with Fic1 (Alexa 647 labeled), Pxl1 (residues 177–188; Alexa 405 labeled), and Cdc12 (residues 20–40; unlabeled). (**B**) 10 µM each of full length Pom1-phosphorylated Cdc15 (E30K, E152K) (Alexa 488 labeled) treated or not with $\lambda$ phosphatase and incubated with Fic1 (Alexa 647 labeled), Pxl1 (residues 177–188; unlabeled), and Cdc12 (residues 20–40; Alexa 405 labeled). Phase separation assays were performed in 50 mM Tris pH 7.4, 250 mM NaCl with 5% PEG. Scale bars, 5 µm.

The online version of this article includes the following video, source data, and figure supplement(s) for figure 9:

**Figure supplement 1.** Purification of full-length Pom1-phosphorylated Flag-Cdc15 (E30K, E152K).

**Figure supplement 1—source data 1.** Raw Coomassie stained gel image with the labeled band of interest.

**Figure 9—video 1.** Apparent fission events of the mNG-Cdc15 condensates in mNG-Cdc15-31A strain shown in *Figure 9C*.
https://elifesciences.org/articles/83062/figures#fig9video1

**Figure 9—video 2.** Apparent fusion events of the mNG-Cdc15 condensates in mNG-Cdc15-31A strain shown in *Figure 9C*.
https://elifesciences.org/articles/83062/figures#fig9video2

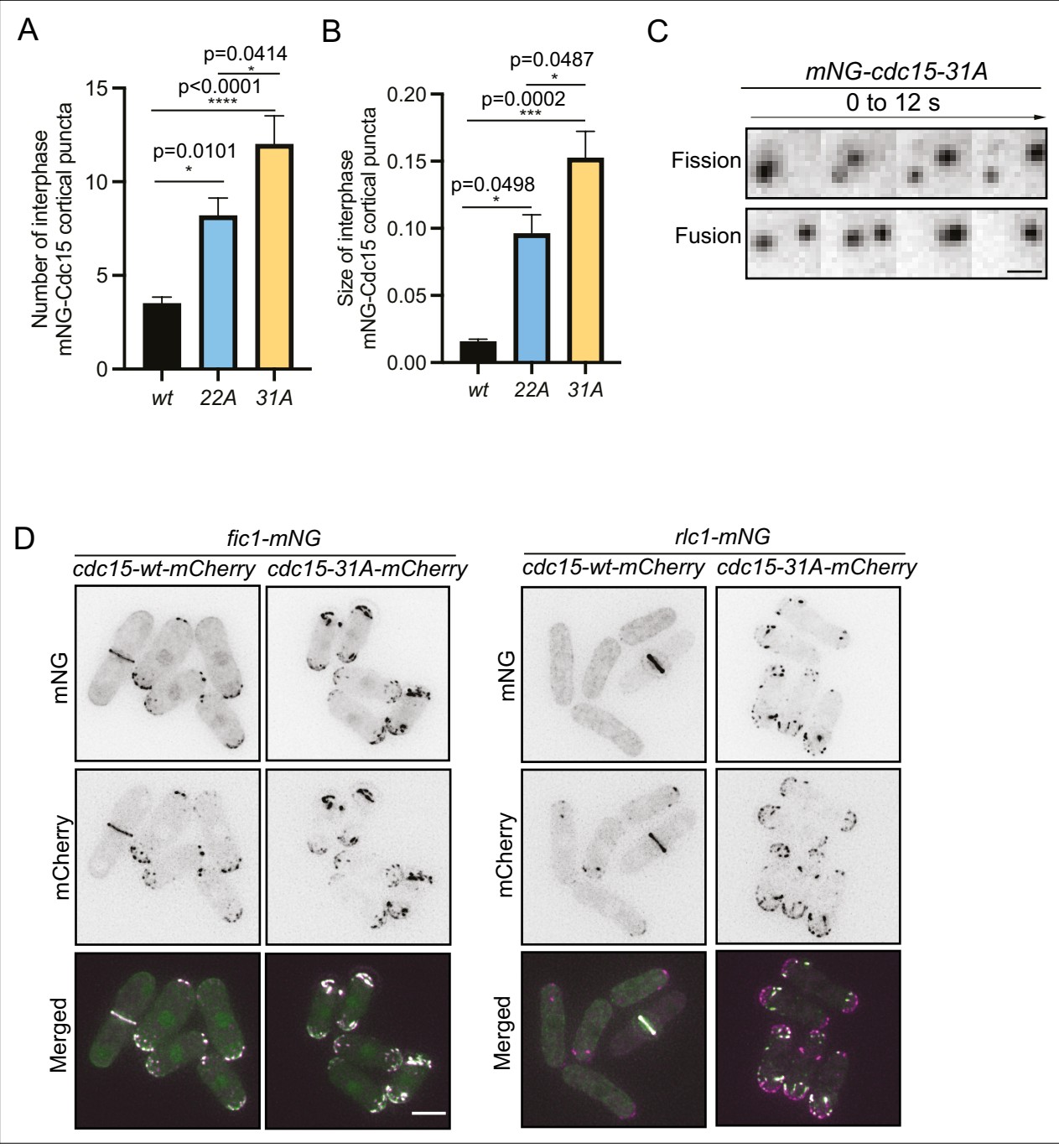

**Figure 10.** Cdc15 assembles into condensates and shows dynamic property in vivo. (**A, B**) Quantification of the number (**A**) and size (**B**) of Cdc15 condensates in the cortex of the indicated strains during interphase. n ≥ 10 cells, N = 3. Error bars indicate SEM. Comparisons were made using one-way ANOVA with Tukey's multiple-comparison test. *p<0.05, **p<0.01, ***p<0.001, and ****p<0.0001. (**C**) mNG-Cdc15 condensates undergoing apparent fission (top) and fusion (bottom) events in *mNG-cdc15-31A* strain. Scale bar, 2 μm. (**D**) Representative images to show Fic1-mNG (left), Rlc1-mNG (right) co-localizing with Cdc15-mCherry from the indicated *cdc15-wt* and *cdc15-31A* strains. Scale bar, 5 μm.

The online version of this article includes the following source data for figure 10:

**Source data 1.** Raw xlsx and pzfx files for *Figure 10A and B* and mov file for *Figure 10C*.

phosphoregulation, particularly given its high intracellular concentration (*Wu and Pollard, 2005*). Pom1, Kin1, and Shk1 share additional substrates involved in cell division (*Lee et al., 2018*; *Magliozzi et al., 2020*), so they may control other CR components in concert with Cdc15 to maximally affect CR position and dynamics. Interestingly, while preventing Cdc15 phosphorylation with the Cdc15-31A

mutant speeds up cytokinesis, consistent with more myosin II being recruited to the CR at an earlier time, our work demonstrates that re-phosphorylation of Cdc15 is not required for CR constriction or cell separation and that Cdc15 need not be removed from the membrane to allow cytokinesis to proceed. Rather, the phosphorylation of Cdc15 helps prevent incorrect division site placement.

We previously hypothesized that Cdc15's conformation was regulated by its phosphostate based on the observations that a site-specific protease cleaved only phosphorylated Cdc15, the phosphorylated and dephosphorylated forms behaved very differently by analytical ultracentrifugation, and only dephosphorylated Cdc15 appeared filamentous by negative-stain EM (*Roberts-Galbraith et al., 2010*). In this work, we studied Cdc15 conformation computationally using coarse-grained molecular dynamics. While such methods lack the detail needed to predict local features such as secondary structure, they are well suited to address phenomena driven by thermodynamic forces representing averages over disordered configurations, such as in LLPS, macromolecular crowding, and relationship between single-chain systems and aggregate behaviors. All-atom simulations provide the highest molecular resolution but suffer from limited sampling when investigating IDR properties even when making use of state-of-the-art methods and computational resources (*Shea et al., 2021*).

Our simulations demonstrate how Cdc15 undergoes a large conformational change upon changing levels of phosphorylation. This behavior reflects the tunability of its IDR, the amino acid sequence of which places it at a transition point between a collapsed globule and an expanded coil. The large size of the IDR and number of phosphorylation sites together with our results using two different coarse-grained potentials (HPS and KH D) indicate that this large reorganization of the IDR is a robust phenomenon. The sharpness of the predicted transition as a function of degree of phosphorylation, however, may depend on uncertainties in coarse-grained interaction potentials: the models used here have not been tuned to reproduce IDR-globular interactions and considered IDRs less than 300 amino acids long, while the Cdc15 IDR is greater than 500 amino acids long. In the HPS model, the stronger pairwise interaction between the IDR and globular region assists in bringing the IDRs together, while the pairwise interactions weakened in the KH D model by SASA scaling leads to weaker IDR-globular interactions and a more abrupt transition (reflected in KH D simulations that are harder to equilibrate). While our simulations have refined our understanding of phospho-regulated Cdc15 conformational changes, other binding interactions of dephosphorylated Cdc15, such as with formin Cdc12, Pxl1, and calcineurin, or itself upon oligomerization (*McDonald et al., 2015*; *Willet et al., 2015a*; *Mangione et al., 2019*; *Bhattacharjee et al., 2020*; *Snider et al., 2020*) may extend or reinforce the conformational changes predicted by simulations of the Cdc15 dimer alone.

Similar to Cdc15 condensation into droplets in vitro, Cdc15 alanine phosphomutants and Cdc15 in strains subjected to kinase inhibition form cortical condensates in vivo, the number and size of which are controlled by the extent of Cdc15 phosphorylation. These puncta also exhibit some liquid-like properties, apparently undergoing fusion and fission events. Further, we observed here and previously (*Roberts-Galbraith and Gould, 2010*) that Cdc15 cortical condensates recruit other CR components, and within the CR, Cdc15 has been observed in node-like structures by super-resolution imaging (*Laplante et al., 2016*; *McDonald et al., 2017*). Taken together, these observations suggest that CR assembly involves condensation of a key membrane-bound scaffold, analogous to what occurs at other actin-based structures such as sites of endocytosis in yeast and focal adhesions in higher eukaryotes (*Day et al., 2021*; *Case et al., 2022*; *Kozak and Kaksonen, 2022*).

In wildtype cells, CRs form from cortical cytokinetic nodes that coalesce into the CR (*Balasubramanian et al., 2004*; *Pollard and Wu, 2010*; *Willet et al., 2015b*; *Rincon and Paoletti, 2016*; *Pollard and O'Shaughnessy, 2019*). Cytokinetic node assembly begins with the anillin-related protein Mid1 arriving at the cortex and becoming stably anchored there (*Wu et al., 2003*; *Wu et al., 2006*; *Clifford et al., 2008*; *Coffman et al., 2009*; *Laporte et al., 2011*; *Akamatsu et al., 2014*). Then, other proteins including Cdc15 are recruited to Mid1 nodes (*Laporte et al., 2011*). Like Cdc15, Mid1 contains a long IDR subject to multisite phosphorylation by a collection of kinases that control its localization (*Bähler et al., 1998*; *Paoletti and Chang, 2000*; *Rincon and Paoletti, 2012*; *Willet et al., 2019*; *Almonacid et al., 2011*) and the Mid1 N-terminus undergoes phase separation to form droplets in vitro (*Chatterjee and Pollard, 2019*). It was proposed that phase separation is not a major mechanism underlying Mid1 node formation because nodes contain fixed component ratios and have not been reported to undergo fusion or fission (*Laplante et al., 2016*; *Chatterjee and Pollard, 2019*). However, Mid1 leaves the CR as it constricts (*Wu et al., 2003*) and CRs can form in the absence of Mid1 (*Chang*

*et al., 1996; Sohrmann et al., 1996; Huang et al., 2007*). FRAP analyses indicate that there are both dynamic and immobile populations of Cdc15 in the CR (*Clifford et al., 2008; Roberts-Galbraith et al., 2009; Roberts-Galbraith and Gould, 2010; Laporte et al., 2011; McDonald et al., 2015; Kamnev et al., 2021*). It is thus possible that initial Cdc15 condensates develop into more stable structures if assembled with Mid1 or other proteins.

Considering our results and the large number of proteins involved in cytokinetic nodes and the CR that involve IDRs with multiple phosphorylation sites, it is intriguing to speculate that phosphorylation pathways drive transitions in nanoscale organization of cytokinetic nodes and contractile ring anchoring by pushing the multicomponent cytokinesis system across condensation, de-mixing, or evaporation phase boundaries (*Jacobs and Frenkel, 2017*), with Cdc15 playing a central role. Interactions with charged lipids, lipid domains on the PM, as well as membrane curvature could also feed into these processes.

## Materials and methods

### Yeast methods

*Supplementary file 1* lists *S. pombe* strains used in this study. Cells were cultured in rich medium YES with supplements or Edinburgh minimal media EMM plus selective supplements (*Moreno et al., 1991*). To integrate *cdc15* mutants at the endogenous locus, *cdc15$^+$/cdc15::ura4$^+$* were transformed with pIRT2-*cdc15* mutant constructs containing 5' and 3' noncoding flanks. LEU+ transformants were sporulated and *cdc15::ura4$^+$*-containing the plasmid were isolated on selective medium. Haploid integrants were then isolated based on their resistance to 5-fluorourotic acid (5-FOA) (United States Biological; F5050) and integration of the *cdc15* mutations was verified by growth on selective media followed by PCR and DNA sequencing. *S. pombe* cells were transformed with a lithium acetate method or by electroporation (*Keeney and Boeke, 1994; Gietz et al., 1995*). Introduction of tagged loci or *cdc15* mutants into other genetic backgrounds was accomplished using standard *S. pombe* mating, sporulation, and tetrad dissection techniques.

To inhibit Pom1$^{as1}$ and Kin1$^{as1}$ in vivo, cells were grown in YES at 32°C to mid-log phase and treated with 4-amino-1-tert-butyl-3-(3-methylbenzyl)pyrazolo[3,4-3]pyrimidine (3MB-PP1) (Toronto Research Chemical; A602960 or Cayman Chemical; 17860) at a final concentration of 15 µM for 30 min. Pck1$^{as2}$ and Shk1$^{as2}$ were inhibited with 30 µM 3-[(3-Bromophenyl) methyl]–1-(1,1-dimethylethyl)–1H-pyrazolo[3,4-d]pyrimidin-4-amine 4-amino-1-tert-butyl-3-(3-bromobenzyl)pyrazolo[3,4-d]pyrimidine (3BrB-PP1) (Abcam; ab143756) for 30 min. Both inhibitors (3MB-PP1 and 3BrB-PP1) were used at the concentrations mentioned above to inhibit combinations of analog-sensitive kinase mutants. As control, cells were grown in equivalent concentration of DMSO (Sigma; D2650).

### Denatured lysis and immunoprecipitation-phosphatase assays

Yeast cell pellets (15 OD) were collected and snap frozen in dry ice-ethanol baths. Pellets were resuspended and washed once in 1 ml NP-40 buffer (6 mM Na$_2$HPO$_4$, 4 mM NaH$_2$PO$_4$, 1% NP-40, 150 mM NaCl, 2 mM EDTA, 50 mM NaF, 4 µg/ml leupeptin, 0.1 mM Na$_3$VO$_4$) with the addition of 1 mM PMSF (Sigma-Alrdrich; P7626), 2 mM benzamidine (Sigma-Aldrich; B6506), and 0.5 mM diisopropyl fluorophosphate (Sigma-Aldrich; D0879-1G). Pellets were then lysed by bead disruption using a Fast-Prep instrument (MP Biomedicals) followed by denaturation by boiling at 95°C for 1 min with 300 µl SDS lysis buffer (10 mM NaPO$_4$, pH 7.0, 0.5% SDS, 1 mM EDTA, 50 mM NaF, 100 µM Na$_3$VO$_4$) with 1 mM dithiothreitol (DTT) and 4 µg/ml leupeptin (Sigma Aldrich;L2884). Lysates were extracted by further addition of 800 µl of NP-40 buffer. After a clearing spin at 13,000 rpm for 5 min at 4°C, the supernatant was collected in a fresh tube and a small portion of the lysate was boiled with 5X SDS sample buffer for immunblotting. For immunoprecipitations, the remainder of the lysates was incubated with anti-Cdc15 polyclonal antibody (VU326) (*Roberts-Galbraith et al., 2009*) for 1 hr at 4°C on a shaker followed by incubation with protein A sepharose (GE Healthcare; 17-5280-04) for 30 min at 4°C. For phosphatase assays, the sepharose beads were washed two times with NP-40 buffer, two times with 1× phosphatase buffer (50 mM HEPES pH 7.4, 150 mM NaCl), and either incubated with buffer control or lambda protein phosphatase ($\lambda$) (New England Biolabs; P0753) according to the manufacturer's protocol. The reactions were stopped by the addition of gel sample buffer. Immunoblot analysis was performed as previously described (*Roberts-Galbraith et al., 2009*). Briefly, to best visualize different

Cdc15 phosphospecies, proteins were resolved in freshly poured (within 24 hr) 8% Tris-glycine gels run at 150 volts for 2.25 hr or pre-poured NuPAGE 3–8% Tris-Acetate gels (Invitrogen; EA03752) run at 150 volts for 2.15 hr (*Figure 1C, D and H*). Then, proteins were transferred to PVDF membrane (Immobilon FL; IPFL00010) for 2 hr. Anti-Cdc15 polyclonal antibody (VU326) or anti-α-tubulin monoclonal antibody (Sigma-Aldrich; B512) was used for immunoblotting. Secondary antibodies were conjugated to IRDye 680 or IRDye 800 (LI-COR Biosciences) and visualized using an Odyssey instrument (LI-COR Biosciences).

## Recombinant protein purification

Bacteria were grown in Terrific Broth media (23.6 g/l Yeast Extract, 11.8 g/l tryptone, 9.4 g/l $K_2HPO_4$, 2.2 g/l $KH_2PO_4$, 4 ml/l glycerol) with appropriate antibiotics to log-phase ($OD_{595}$ 1–1.5) at 36°C. Cells were incubated on ice for 15 min and protein expression was initiated with the addition of 0.4 mM isopropyl β-D-1-thiogalactopyranoside (IPTG) (Fisher Scientific; BP1755). Cells were then incubated for 16–18 hr at 18°C for optimal protein production.

To purify MBP and GST fusion proteins, frozen cell pellets were lysed in either MBP buffer (20 mM Tris-HCl pH 7.4, 150 mM NaCl, 1 mM EDTA, 1 mM DTT) or GST buffer (4.3 mM $NaHPO_4$, 137 mM NaCl, 2.7 mM KCl, 1 mM DTT) with the addition of 200 µg/ml lysozyme (Sigma-Aldrich; L6876), cOmplete EDTA-free protease inhibitor cocktail (Roche; 05056489001), and 0.1% NP-40 (US Biologicals; N3500). Buffers were modified for MBP-Pxl1 to exclude EDTA. Continuous agitation on ice for 20 min was used to suspend the cell pellet. Then, lysates were sonicated three times for 30 s, with at least a 30 s pause between sonications (Sonic Dismembrator Model F60, Fisher Scientific; power 15 watts). Lysates were cleared for 15–30 min at 10–13K rpm. Cleared lysate was then used in a batch purification protocol by addition of either amylose (New England Biolabs, Inc; E8021L) or GST-bind (EMD Millipore; 70541) resin for 2 hr at 4°C. Then, resin was washed three times for 5 min at 4°C with the appropriate buffer. To elute proteins, resins were resuspended in an equal volume of appropriate elution buffer and mixed for 30 min at 4°C. MBP fusion proteins were eluted in MBP buffer supplemented with 10 mM maltose (Fisher Scientific; M75-100) and GST proteins were eluted in GST elution buffer (50 mM Tris-HCl pH 8, 10 mM glutathione; Sigma-Aldrich; G4251). The supernatant was separated from the resin to a fresh tube. Eluted fusion proteins were then aliquoted, snap frozen, and stored at –80°C. GST-Cdc15C-19A was made by mutating the Pom1 phosphorylation sites (T419, T492, S511, S527, S605, S636, S702, S710, S721, S732, S743, S752, S774, S785, S786, S813, S821, S831, S836, S840) to alanine in the GST-Cdc15C vector (*Bhattacharjee et al., 2020*).

Full-length Cdc15 oligomerization mutant (Cdc15-E30K, E152K), co-produced in bacteria with Pom1, was prepared as described previously (*Bhattacharjee et al., 2020*). To purify $His_6$-Cdc15-IDR-SH3 (residues 327–927), $His_6$-Cdc15-IDR-SH3 (residues 327–927) (W903S), His-Fic1 or His-mCherry, frozen cell pellets were lysed in buffer (50 mM Tris-HCl pH 7.4, 500 mM NaCl, 1 mM TCEP) with the addition of 200 µg/ml lysozyme (Sigma-Aldrich; L6876), cOmplete EDTA-free protease inhibitor cocktail (Roche; 05056489001), and 0.1% NP-40 (US Biologicals; N3500). Then, lysates were sonicated three times for 30 s, with at least a 30 s pause between sonications (Sonic Dismembrator Model F60, Fisher Scientific; power 15 watts). Lysates were cleared for 15–30 min at 10–13K rpm. Cleared lysate was then applied to His-Trap HP (1 ml) (Cytiva; 17524701) using an AKTA pure system (Cytiva). Proteins bound to the column were eluted using freshly prepared buffer (50 mM Tris-HCl pH 7.4, 500 mM NaCl, 1 mM TCEP and 200 mM imidazole) and then concentrated using an Amicon Ultra 0.5 Centrifugal Unit (MiliporeSigma; UFC501096).

Protein concentrations were calculated by either comparison of Coomassie Brilliant Blue G (CB) (Sigma-Aldrich; B0770) stained SDS-PAGE-separated purified proteins to bovine serum albumin (BSA) standards (Sigma-Aldrich; A2153) by using a NanoDrop spectrophotometer (ND1000) and/or by using Pierce BCA Protein Assay Kit (Thermo Scientific; 23225).

## In vitro kinase assays

Radioactive in vitro kinase assays were performed in Kin1 buffer (50 mM Tris pH 7.5, 25 mM NaCl, 10 mM $MgCl_2$, 1 mM DTT, 200 µM ATP), Shk1 buffer (40 mM HEPES, pH 7.5, 10 mM $MgCl_2$ with 200 µM ATP), or Pck1 buffer (4 mM HEPES, pH 7.5, 5 mM $MgCl_2$, 2 mM DTT, 0.5 mM EGTA with 200 µM ATP) with 4 µCi γ-$^{32}$P-ATP (PerkinElmer BLU002250UC). For each 25 µl kinase reaction, 0.4 µg of kinase (MBP-Kin1 [amino acids 1–520,T299D], GST-Shk1 and MBP-Pck1 [amino acids 644–988,T823E]) and

2 μg of substrate (MBP, MBP-Cdc15N [amino acids 1–405], MBP-Cdc15C [amino acids 441–927]) were used. After 30 min at 30°C, reactions were stopped with the addition of 10 μl 5× sample buffer, boiled, and analyzed by SDS-PAGE. Inputs were detected by CB staining, and $^{32}$P incorporation was detected by autoradiography.

For testing the effect of one kinase on the phosphorylation of Cdc15C by another, in vitro kinase assays were performed first with unlabeled ATP. The substrates (MBP-Cdc15C or GST-Cdc15C) were on beads and the kinases were in the supernatant. After the first kinase assay (30 min at 30°C), the beads were washed three times in 500 μl of the corresponding 1X kinase assay buffer to remove the kinase. Bead-bound substrates (phosphorylated or not in the first reaction) were then divided into separate tubes and incubated with one of the three kinases or buffer control in the presence of 4 μCi γ-$^{32}$P-ATP. All secondary kinase assays were performed in 25 μl total volume for 30 min at 30°C. These reactions were stopped with the addition of 10 μl 5X sample buffer, boiled for 2 min, and analyzed by SDS-PAGE. Inputs were detected by CB staining, and $^{32}$P incorporation was detected by autoradiography.

Kinase reactions for phosphopeptide mapping were performed with 0.2 μg of each kinase added to 2 μg substrates, GST-Cdc15C (amino acids 441–927), or GST-Cdc15C1 (amino acids 600–927) in its corresponding kinase assay buffers with 100 μM cold ATP, 4 μCi γ-$^{32}$P-ATP, and 10 mM MgCl$_2$ at 30°C for 30 min. Reactions were quenched by boiling in SDS-PAGE sample buffer.

Kinase assays for the identification of phosphorylation sites were performed with 200 μM unlabeled ATP, 0.2 μg of recombinant kinase, and in 25 μL final volume. After a 15 min incubation with kinase, a second equal aliquot of kinase was added and the reaction stopped after and additional 45 min. In each case, 2 μg of GST-Cdc15C was used as substrate.

## Phospho-peptide mapping

Phosphorylated proteins (GST-Cdc15C and GST-Cdc15C1) were resolved by SDS-PAGE, transferred to PVDF membranes, and visualized by autoradiography. Phosphopeptides were released from membrane slices with 10 μg trypsin at 37°C overnight with a second addition of 10 μg trypsin and incubation at 37°C the next day for 2 hr. Released peptides were lyophilized and resuspended in pH 1.9 buffer (*Boyle et al., 1991*). Tryptic phosphopeptides were separated in the first dimension by thin-layer electrophoresis and in the second dimension by chromatography (*Boyle et al., 1991*). TLC plates were exposed to film for 2–4 days with intensifying screens.

## Phosphorylation site identification by mass spectrometry

TCA-precipitated proteins were digested and analyzed by two-dimensional liquid chromatography/ tandem mass spectrometry as described previously (*Chen et al., 2013*), with the following modifications: proteins were digested by trypsin, chymotrypsin, and elastase and the number of salt elution steps was reduced to 6 (i.e., 0, 25, 50, 100, 600, 1000, and 5000 mM ammonium acetate). Peptide identifications were filtered and assembled using Scaffold (version 4.8.4; Proteome Software) and phosphorylation sites were analyzed using Scaffold PTM (version 3.1.0) using the following filters: minimum of 99% protein identification probability, minimum of five unique peptides, and minimum of 95% peptide identification probability.

## Microscopy and image analysis

Yeast for live-cell imaging were grown at 25°C. Live-cell images of *S. pombe* cells were acquired using a Zeiss Axio Observer inverted epifluorescence microscope with Zeiss 63× Oil (1.46 NA) and captured using Zeiss ZEN 3.0 (Blue edition) software. The acquisition setting for mNG-Cdc15 was set at 95% (intensity) and 600 ms (exposure time) with a Z-stack step size of 0.25 um and a total of 21 Z-slices.

To quantify the ratio of mNG-Cdc15 on the PM to mNG-Cdc15 in the cytoplasm of interphase cells, we used two homemade scripts (macros). The first macro subtracted the background and sum projects the slices. A second macro utilized the '3D Object Counter' plugin (*Bolte and Cordelières, 2006*) to measure 3D-object fluorescence intensity of mNG-Cdc15 assemblies along with mNG-Cdc15 total cell fluorescence. The settings for the 3D Object Counter plugin were as follows: the intensity threshold was set to 500, the minimum voxel size was 5, and the maximum size was set as 20,000. The cytoplasmic signal was calculated by subtracting the 3D object intensity from whole-cell intensity. The 3D object intensity was then divided by the cytoplasmic signal to generate the final ratio.

Quantifications of Cdc15 condensate size and number in cells were performed on deconvolved, max-projected grayscale images. Maximal and minimal threshold pixel intensities were fixed using *cdc15-31A* cells and those thresholds were made constant through the analyses of all strains. Within Image > Adjust > Threshold, the dropdown settings selected were 'Max entropy' and 'B&W.' After saving the settings, the images were changed to binary mode for final analysis of the area (size) and total number of condensates from each selected image.

Time-lapse imaging was performed on log-phase cells at 25°C using a Personal DeltaVision (Cytiva Life Sciences, Marlborough, MA) that includes a microscope (IX71; Olympus), 60× NA 1.42 Plan Apochromat and 100× NA 1.40 U Plan S Apochromat objectives, fixed and live-cell filter wheels, a camera (a pco.edge 4.2 sCMOS), and softWoRx imaging software (Leica). Time-lapse imaging was performed in YES media in a CellASIC ONIX microfluidics perfusion system (MilliporeSigma, Burlington, MA). Cells were loaded into Y04C plates for 5 s at 8 psi, and YES liquid medium flowed into the chamber at 5 psi throughout the time-lapse. Images were acquired every 2 min with 0.5 μm optical spacing. Time-lapse images were deconvolved with 10 iterations and visualized as maximum projections. Images used for fluorescence quantification were not deconvolved. Total internal reflection fluorescence (TIRF) images were also recorded on the Personal DeltaVision described above. The 488 laser was used for illumination of Alexa-488 labeled Cdc15-IDR-SH3.

Quantitative analysis of microscopy data was performed using Fiji (a version of ImageJ software available at https://fiji.sc; *Schindelin et al., 2012*). All quantifications were performed on non-deconvolved, sum projected images. For all intensity measurements, the background intensity was subtracted. The background intensity was determined by taking a measurement in an area of the image without cells. The raw intensity of the background was divided by its area, which was multiplied by the area of the intensity measurement of interest. This number was subtracted from the raw integrated intensity measurement of interest (*Waters, 2009*). For *Figure 4A and B*, mNG-Cdc15 and Rlc1-mCherry intensities in the CRs were normalized to their respective whole-cell intensities.

## Quantification of incorporation index in in vitro LLPS assay

To quantify the homogeneity of the droplets, line scans of 1 pixel each were obtained from just outside one side of the droplet through to the other, crossing through the precise middle of each droplet. The fluorescence intensities of the middle ($X_2$) and edge (just prior to intensity drop at the droplet edges) ($X_1$) were determined. The incorporation index was calculated as the ratio of $X_2/X_1$.

## Quantification of tip septa

Tip septa were quantified using a previously published protocol (*Bhattacharjee et al., 2020*).

## Protein fluorescent labeling

Purified recombinant proteins were labeled using amine reactive NHS ester dyes (Thermo Fisher Scientific). The Alexa fluorophores used to label proteins were Cdc15 – Alexa 488 (A-20000, Thermo Fisher); Fic1 – Alexa 647 (A-20006, Thermo Fisher); Pxl1 peptide (residues 177–189) – Alexa 405 (A30000, Thermo Fisher); and Cdc12 peptide (residues 20–40) – Alexa 405 (A30000, Thermo Fisher). All proteins and peptide were in 50 mM Tris, pH 7.4, 500 mM NaCl, 1 mM TCEP before the addition of dye. After addition of the dyes (dyes: protein molar ratio was 1:1), solutions were incubated for 1 hr on ice and excess dye was removed using Pierce Zeba Desalt Spin Columns (Thermo Scientific; 89882). The labeling efficiency was determined by measuring the protein concentration using Pierce BCA Protein Assay Kit (Thermo Scientific; 23225) before and after labeling with dyes.

## In vitro LLPS assays

For all in vitro phase separation assays, Cdc15-IDR-SH3, His-Cdc15-IDR-SH3, or full-length Cdc15-E30K, E152K were co-expressed in bacteria with Pom1 (*Bhattacharjee et al., 2020*), purified as described above or previously in the case of full-length Cdc15-E30K, E152K (*Bhattacharjee et al., 2020*), and labeled with fluorescent dye as described above. Phase separation was induced by treatment with $\lambda$-phosphatase (NEB; P0753L) and dilution to physiological salt concentration (50 mM Tris pH 7.4, 150 mM NaCl final) in the presence or absence of a crowding agent (5% PEG, average molecular weight 3350; Sigma; P4338). For solution assays, the resultant Cdc15-IDR-SH3 concentration ranged from 5 to 50 μM and 5–10 μl was added to a glass coverslip that had been thoroughly cleaned

with isopropanol and imaged on the DeltaVision microscope system described above. Droplets were considered to be phase separated if they showed liquid-like properties (visible rapid fusion events) and were dynamic (rapid recovery in a FRAP assay). Phase diagrams at different concentrations of salt and protein were determined within 5 min of phosphatase treatment.

For LLPS assays of Cdc15-IDR-SH3 with Fic1 and/or Pxl1 peptide in solution, 10 µM of each were mixed and the final salt concentration was diluted to 150 mM NaCl in the presence of 5% PEG before addition of $\lambda$-phosphatase. Full-length Cdc15-E30K,E152K, Fic1, Pxl1 peptide, and Cdc12 peptide were combined at 10 µM and diluted to a final concentration of 250 mM NaCl in the presence of 5% PEG before the addition of $\lambda$-phosphatase. Images of droplets were acquired within 5 min of $\lambda$-phosphatase addition.

For LLPS assays on membrane, supported lipid bilayers were generated as previously described (*Banjade and Rosen, 2014*; *Day et al., 2021*). A mixture of 98% DOPC (Avantilipids; 850375P-25mg), 2% DGS-NTA-Ni (Avantilipids;790404P-5mg), and 0.1% 18:1 PEG5000 PE (Avantilipids; 880230P-25mg) was evaporated under a nitrogen gas stream and then dried under vacuum for 1 hr. The resultant pellet was hydrated in lipid buffer (25 mM HEPES, pH 7.5, 150 mM NaCl, 1 mM MgCl$_2$, 1 mM TCEP) for 30 min, vortexed gently, and then frozen and thawed 10 times. The liposome sample was passed through a filter with an extruder 10 times to obtain small unilamellar vesicles (SUVs). Prior to the experiment, coverslips and glass slides were thoroughly washed with 6 M NaOH and water followed by 70% ethanol. Clean glass slides and cover slips were then equilibrated in lipid buffer. A homemade chamber was constructed using the coverslips, double-sided tape, and glass slides. 10–15 µl of SUVs were added to the chamber and incubated for 30 min at room temperature to allow SUVs to settle onto the glass and fuse to form the bilayer. Bilayers were washed three times with lipid buffer and blocked with lipid buffer containing 0.1% BSA for 30 min. Then, fluorescently labeled His-Cdc15-IDR was added in the range of 5–50 µM to the bilayer and incubated at room temperature for 1 hr. The slides were washed with lipid buffer containing 0.1% BSA to remove unbound His-tagged proteins. Fluorescently labeled His-Cdc15-IDR bound to the lipid bilayer was then visualized by TIRF microscopy.

## Fluorescence recovery after photobleaching (FRAP)

FRAP analysis of droplets that were between 5 µm and 10 µm diameter was performed on the Personal DeltaVision microscope described above. A portion of a droplet (~1–2 µm) was photobleached with a 488 laser. Images were acquired every 1 s for 60 s and five images were taken prior to the photobleaching event. Quantifications of the fluorescence recovery were performed using ImageJ. FRAP intensity measurements were corrected for background and time-course photobleaching. Each FRAP measurement was obtained from a distinct droplet, and a minimum of 10 droplets was analyzed in three separate experiments.

## Coarse-grained molecular dynamics simulations

Each amino acid is represented as a single bead at the location of the α carbon. Simulations were performed in either the HPS model (*Dignon et al., 2018*; *Perdikari et al., 2021*) or the KH model D (*Kim et al., 2008*; *Dignon et al., 2018*) in LAMMPS (*Plimpton, 1995*) using parameters described in the respective model papers unless otherwise stated. In each case, pairwise interactions are modeled by Lennard–Jones-type potentials with a 3σ cutoff. We used an electrostatic screening length of 10 Å, and uniform dielectric constant 80 for electrostatic interactions of charged residues. Pairwise interactions for phosphorylated resides for the HPS model are as in *Perdikari et al., 2021*. For the KH model that has not been parameterized for phosphorylation, phosphorylated serine and threonine are treated as aspartic acid pair interactions, while charge and interaction radii are updated from *Perdikari et al., 2021* to match the HPS model.

We represent the known structure of the F-BAR dimer (PDB 6XJ1; *Snider et al., 2020*) and the Alphafold predicted structure of the Cdc15 SH3 (AF-Q09822-F1; *Jumper et al., 2021*) as rigid bodies using the fix rigid command in LAMMPS. All other residues of the Cdc15 sequence are represented as beads connected to their neighbors with harmonic bonds with a 3.81 Å equilibrium length and a 378 kcal/(mol Å$^2$) spring constant. For all HPS model simulations and for the KH model single-chain IDR simulations, we implemented the non-bonded pairwise potential using continuous piecewise LAMMPS potentials as in *Smith et al., 2021* (code available at https://github.com/aah217/KH_

LAMMPS and https://github.com/aah217/HPS_LAMMPS). We performed these simulations with the following versions of LAMMPS: '18 Apr 2015,' '21 Jul 2020,' '29 Oct 2020,' and '10 Mar 2021.' As the KH model D scales non-bonded pairwise interactions based on the solvent accessible surface area (SASA) of a residue relative to a free chain, we implemented this functionality in LAMMPS version '29 Sep 2021 - Update 1' as a custom potential utilizing code provided by Jeetain Mittal. We set the SASA-based value to 1.0 for all residues not in rigid bodies (*Dignon et al., 2018*), and as calculated at http://curie.utmb.edu/getarea.html (*Fraczkiewicz and Braun, 1998*) for residues in the F-BAR or SH3 structures. Integration was performed using a 10 fs timestep with neighbor lists updated every 10 steps. Simulations were carried out in the NVT ensemble through the use of the NVE integrator and a Langevin thermostat with a 1 ps$^{-1}$ frictional coefficient.

IDR simulations were initialized in one of two ways. For the dephosphorylated case using the HPS model, the system was started in a straight-line configuration in which the amino acids were spaced 3.81 Å apart on one axis. All other simulations were started using a semi-relaxed configuration of the IDR with an $R_G$ of 6.53 nm. Individual trajectories were run long enough for $R_G$ to fluctuate around an average value, with individual trajectory durations ranging from 500 ns to greater than 10 μs, with higher temperature simulations running for longer times due to computational efficiency, and between 6 and 8 trajectories for each case. For trajectories that started from a specified initial condition, as opposed to continuations, an autocorrelation function was fit to the $R_G$ versus time data to calculate the relaxation time. Data up to twice the relaxation time were excluded from calculating the measured $R_G$ values calculated and reported. All remaining $R_G$ versus time data for a particular case were split into five equal parts, the mean calculated for each part, and the standard deviation of the five means is reported as the error bars in *Figure 6F*.

To initialize full dimer simulations, two straight-line IDRs were placed with the amino acid next in sequence from the F-BAR dimer placed 3.81 Å from its neighbor on the F-BAR dimer with IDRs extending away from one another. Each bead in the IDR was spaced 3.81 Å apart. An SH3 domain was connected at the end of each IDR. A first pass relaxation was performed in LAMMPS for 10 ns with the spring constant reduced to 0.378 kcal/(mol Å$^2$) and using the fix deform command to shrink the box size, and the relative spacing of its contents (excluding those fixed rigid), by a factor of 10. Follow-up simulations of 10 ns were performed to progressively increase the spring constant back to that defined by the model, making use of the nve/limit command in LAMMPs, which imposes a maximum distance that beads can travel from one timestep to the next. This method of relaxation resulted in the configuration used in the IDRs started apart simulations (*Figure 6A*, *Figure 6—figure supplement 1A*). For the IDRs interacting initial condition (*Figure 6—figure supplement 1B*), a short simulation was performed following from the previous steps in which a force of sufficient strength was applied continuously to each SH3 in the direction of the opposite side of the F-BAR, causing the SH3s to cross the F-BAR dimer midplane. Follow-up simulations were performed to relax the IDRs in the new configuration using steps similar to the IDRs-apart relaxation. For the reported IDR COM separation data, an autocorrelation function was fit to calculate the relaxation time away from the initial condition and data up to twice the relaxation time was not included.

## Statistical analysis

Calculations of mean, standard error of the mean (SEM), and statistical significance were performed with Prism 6.0 or Prism 8.0 (GraphPad Software). Sample size (n), replicates (N), and statistical test are included in figure graphs or legends. For image analyses, no raw data were excluded with the exception of cells that were not in focus or cells that moved during imaging.

## Acknowledgements

We are grateful to Nathan McDonald for advice on LLPS assays, Mohan Balasubramanian, Juraj Gregan, and Xavier Le Goff for providing strains, and Jitender Singh Bisht for performing the experiments shown in *Figure 5A* and *Figure 5—figure supplement 1A*. We thank Jeetain Mittal for providing feedback on the simulations and source code for the KH D model implemented in LAMMPS. Portions of this research were conducted on Lehigh University's Research Computing infrastructure partially supported by NSF Award 2019035 and the high-performance computing capabilities of the Extreme Science and Engineering Discovery Environment (XSEDE), supported by the National Science Foundation, project no. TG-MCB180021. MCM was supported by NIH grants F31GM119252

and T32GM007347. This work was supported by NIH grant R35GM136372 to DV and NIH grant R35GM131799 to KLG.

## Additional information

### Funding

| Funder | Grant reference number | Author |
|---|---|---|
| National Institutes of Health | R35GM136372 | Dimitrios Vavylonis |
| National Institutes of Health | R35GM131799 | Kathleen L Gould |
| National Institutes of Health | F31GM119252 | MariaSanta C Mangione |
| National Institutes of Health | T32GM007347 | MariaSanta C Mangione |
| National Science Foundation | 2019035 | Dimitrios Vavylonis |
| National Science Foundation | TG-MCB180021 | Dimitrios Vavylonis |

The funders had no role in study design, data collection and interpretation, or the decision to submit the work for publication.

### Author contributions

Rahul Bhattacharjee, Conceptualization, Data curation, Formal analysis, Validation, Investigation, Writing – original draft, Writing – review and editing; Aaron R Hall, Conceptualization, Data curation, Formal analysis, Investigation, Visualization, Writing – original draft, Writing – review and editing; MariaSanta C Mangione, Conceptualization, Formal analysis, Investigation, Visualization, Writing – original draft, Writing – review and editing; Maya G Igarashi, Formal analysis, Investigation, Visualization, Writing – original draft, Writing – review and editing; Rachel H Roberts-Galbraith, Formal analysis, Investigation, Visualization, Writing – review and editing; Jun-Song Chen, Data curation, Formal analysis, Validation, Investigation, Visualization, Writing – original draft, Writing – review and editing; Dimitrios Vavylonis, Conceptualization, Resources, Data curation, Formal analysis, Supervision, Funding acquisition, Validation, Investigation, Visualization, Writing – original draft, Project administration, Writing – review and editing; Kathleen L Gould, Conceptualization, Resources, Formal analysis, Supervision, Funding acquisition, Validation, Investigation, Writing – original draft, Project administration, Writing – review and editing

### Author ORCIDs

Rahul Bhattacharjee ![ORCID] http://orcid.org/0000-0001-8607-5756
Aaron R Hall ![ORCID] http://orcid.org/0000-0002-6194-8060
Maya G Igarashi ![ORCID] http://orcid.org/0000-0003-3727-5473
Rachel H Roberts-Galbraith ![ORCID] http://orcid.org/0000-0002-2682-2366
Dimitrios Vavylonis ![ORCID] http://orcid.org/0000-0003-1802-3262
Kathleen L Gould ![ORCID] http://orcid.org/0000-0002-3810-4070

### Decision letter and Author response

Decision letter https://doi.org/10.7554/eLife.83062.sa1
Author response https://doi.org/10.7554/eLife.83062.sa2

## Additional files

### Supplementary files
• Supplementary file 1. *S. pombe* strains used in this study.

- Supplementary file 2. DNA oligos and peptides used for this study.
- Supplementary file 3. Protein concentration measurement data using BCA method.
- MDAR checklist

### Data availability

All data generated or analyzed during this study are included in the manuscript and supporting file; Source Data files have been provided for Main Figures 1,2, 3, 4, 5, 6, 7, 10, Figure 2-figure supplement 1, Figure 3-figure supplement 1-3, Figure 5-figure supplement 2, Figure 6-figure supplement 1-5, Figure 7-figure supplement 1, Figure 8-figure supplement 1 and Figure 9-figure supplement 1. Files corresponding to our molecular dynamics simulations (input files, initial system state, and python analysis scripts) are available at https://github.com/aah217/Cdc15_CGMD (copy archived at swh:1:rev:56beb77cf945848fdd9d4de389073b333ce0e876). Because of the very large size of trajectory files, these files will be available permanently without restrictions from D. Vavylonis upon simple request, according to eLife's policies.

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
