## [Editor Report]

Yeast and humans use an actin-myosin II-based division apparatus, which generates forces for cell division. A key factor that regulates cell division apparatus assembly and ensures its stability is Cdc15. The authors use a combination of biochemistry, advanced imaging, and genetics to elucidate how protein kinases, i.e., molecules that attach signaling phosphate moieties to client proteins, regulate the formation of 'condensates' of Cdc15, and how this modulates the correct assembly of the cell division apparatus.

---

## [Decision Letter]

**Decision letter after peer review:**

Thank you for submitting your article "Multiple polarity kinases inhibit phase separation of F-BAR protein Cdc15 and antagonize cytokinetic ring assembly in fission yeast" for consideration by *eLife*. Your article has been reviewed by 2 peer reviewers, and the evaluation has been overseen by a Reviewing Editor (Mohan Balasubramanian) and Anna Akhmanova as the Senior Editor. The reviewers have opted to remain anonymous.

The 2 referees (referred to as referees 1 and 2) are very supportive but have raised a number of points for clarification and I would like you to consider and fix as many as possible since these are largely about data presentation and improving clarity.

The two key experiments that I see as essential are mentioned below.

Essential revisions:

1. As suggested above, phase separation of purified Cdc15-31A will provide strong evidence that the sites identified are physiologically involved in phase separation. Moreover, it would be useful to show if these mutants undergoing phase separation also recruit cytokinetic proteins.

2. In figure 4 the authors show that Cdc15-31A mutants prolong ring formation and decrease the time of ring constriction. They also claim that the time of ring maturation is decreased. However, it is well documented that altering the time of ring formation can inversely alter the time of ring maturation since ring maturation depends on the cells' ability to form a septum. A better way to describe the changes in the timing of cytokinetic events is to use the duplication of the spindle pole body marker as the starting event instead of using the completion of ring formation as a marker given that ring formation is itself impacted.

*Reviewer #1 (Recommendations for the authors):*

Figure 1

Some of the immunoblotting shown in this manuscript is not clear in demonstrating the phosphorylation status of Cdc15. Due to the technical limitation, it is understandable. Comparing the IP samples and lysate sample (only shown in B), I assume the lysate blot seems better to demonstrate the mobility shift (presumably due to dephosphorylation or degradation during IP?). Could the authors show F and G using the lysate samples to improve the quality of the blot?

The mobility is less evident in the blots, so they should be enlarged in the figure panels to help readers. I recommend that this figure should only show the enlarged blot of analog-sensitive (AS) mutants, and the panels for temperature-sensitive or deletion mutants should be moved to supplemental figure 1.

Panel (A) is the same experiment as the left two lanes of the panel (C), so this could be excluded from the figure or moved to supplemental figure 1.

There are multiple bands in the blots. It is not clear which bands correspond to Cdc15/phosphor-Cdc15. Could the author indicate the target bands?

The bottom bands are not shown correctly in panel (B) α-tubulin blot. The author should show which bands correspond to α-tubulin used as the loading control of the experiment. If the bottom bands correspond to α-tubulin, this blot has to be replaced with the one showing all the bands clearly.

For the AS mutant experiments, it would be better to ensure that the kinase activity is inhibited. The authors may want to show the phosphorylation of a kinase substrate in the lysate blot as a lead-out of the kinase activity.

Figure1-supplement 1

This supplemental material seems not supplemental material for Figure 1. I assume this is supported information for Figure 2.

(A) In CB gel images, it is not clear which band corresponds to MBP/MBP-Cdc15N/MBP-Cdc15C. I recommend that the authors indicate them using arrowheads.

(B) showed that the 19D mutant got less phosphorylation by Shk1/Pck1 than the 19A mutant. This suggests that Pom1-mediated phosphorylation inhibits Shk1/Pck1 to phosphorylate Cdc15. This supports the data shown in Figure 2, but the data is not discussed in the main text. The author should discuss the results if they include this result in this paper.

Figure 2

It may be helpful for readers to see schematics of the design of Cdc15C truncations.

Figure 4

I recommend that authors replace the bar graph with a swarm plot or violin plot, which gives readers variance in the data.

(A and B) The intensity of the Cdc15-mNG on the ring can be normalized by whole cell intensity.

In the right panel of A, the intensity can be plotted against the ring diameter.

(C) The same as A, the intensity of Rlc1-mCherry on the contractile ring can be normalized by whole cell intensity. For the right panel of C, the fluorescent intensity of the ring should be plotted against the ring diameter.

Figure 5

I recommend that authors replace the bar graph with a swarm plot or violin plot, which gives readers variance in the data.

Figure 7

(F) It is unclear how authors quantify the fluorescent intensity of the edge and center of condensate. The author should show an example ROI for the measurement. I assume this was conducted by line plot or particle analysis.

(G and H) I assume the authors used His-tag and Ni-NTA interaction to anchor the His-IDR-SH3 on SLB. Although the BAR domain is not present in this construct, this must be clearly stated in the main text and figure legend to avoid confusion because Cdc15 is a well-known membrane-bound protein.

(G) should be His-Cdc15-IDR-SH3 rather than Cdc15-IDR-SH3.

Figure 8

(A) the bottom panel, the condensate shows the non-uniform distribution of the fluorescent, which is inconsistent with the author's conclusion from Figures 7 E and F. Does it because of the presence of other proteins? I see some patterns on the condensates.

(B) The author stated that "we observed that 84% of droplets contained at least three" in line 361 in the main text. Could they show the ratio in the main figure? This should be discussed carefully in the manuscript. In my understanding, the author reconstitutes the Cdc15 condensation using equimolar protein concentration. It would be interesting to conduct the same experiment under the physiological molar ratio of these proteins.

Figure 8 -supplemental 2

I cannot see the images. All panels I have downloaded are black boxes.

Figure 9

(A and B) The author should use a violin plot or swarm plot instead of the bar graph.

(C) could they also conduct FRAP in cells? Showing recovery could prove the LLPS rather than protein aggregation.

Discussion

Line 485. One of the reference formats is not consistent with others.

Materials and Method

Recombinant protein purification

Line 603 – 606

The methodology of the protein concentration measurement should be shown individually. If the measurement is conducted in inconsistent ways, the author cannot prove the equimolar concentration in their experiments. And I have a little concern about their protein preparation. There are many degraded or contaminated proteins in the protein used in the experiments. In such a case, the concentration measurement using Nanodrop (absorbance of 280 nm light) may not be reliable.

Quantification of mNG-Cdc15 PM to cytoplasmic ratio

Line 695 – 697

This section should have more description of the 2nd macro they used. The parameter from the 3D Object Counter plugin used for the calculation was not clearly stated in the section. In this way, it is difficult for the reader to understand and reproduce the experimental design in this study.

Protein fluorescent labelling

Line 712 – 721

The author should show how they calculated the concentration of labelled proteins. If they use Nanodrop, the protein sample should not have degradations and contaminants and the absorbance of 280 nm light on the fluorophore must be subtracted for the calculation.

Using amin-reactive dyes, labelling efficiency cannot be measured properly if there are multiple Lys in the protein. Exposed Lys residues would be labeling sites and it is difficult to control the labelling ratio in such cases.

*Reviewer #2 (Recommendations for the authors):*

This is a very well-written paper that provides a deep mechanistic understanding of how phosphorylation of intrinsically disordered regions can prevent phase separation and consequent protein function. This paper is very well suited for *eLife*. I have some suggestions mentioned below that will further strengthen the paper.

1. As suggested above, phase separation of purified Cdc15-31A will provide strong evidence that the sites identified are physiologically involved in phase separation. Moreover, it would be useful to show if these mutants undergoing phase separation also recruit cytokinetic proteins.

2. In figure 4 the authors show that Cdc15-31A mutants prolong ring formation and decrease the time of ring constriction. They also claim that the time of ring maturation is decreased. However, it is well documented that altering the time of ring formation can inversely alter the time of ring maturation since ring maturation depends on the cells' ability to form a septum. A better way to describe the changes in the timing of cytokinetic events is to use the duplication of the spindle pole body marker as the starting event instead of using the completion of ring formation as a marker given that ring formation is itself impacted.

3. Also, Figure 8—figure supplement 2 is very hard to follow. The figure is too dark and the signal contrast is barely visible but upon zooming in, I was able to see dots. Inverting the image would help with the lack of contrast. This figure also misses a scale bar.

---

## [Author Response]

Essential revision:1. As suggested above, phase separation of purified Cdc15-31A will provide strong evidence that the sites identified are physiologically involved in phase separation. Moreover, it would be useful to show if these mutants undergoing phase separation also recruit cytokinetic proteins.

This concern seems to be based on two misunderstandings. First, the method of producing and purifying phosphorylated Cdc15 was not well-described in the first version of the manuscript. This has been corrected. To elaborate, in order to purify Cdc15 from bacteria for LLPS assays, it must be co-produced with one of its kinases. Otherwise, it is not phosphorylated and it is not soluble. Thus, the proposed experiment of using recombinant Cdc15-31A is technically not possible. For the experiments presented in this paper, we co-produced Cdc15 with Pom1 to obtain the phosphorylated form of Cdc15. Pom1 phosphorylates 22 of the 31 sites mutated in the 31A mutant; these sites were identified in previous work and they are physiologically relevant in vivo (Bhattacharjee et al., 2020). Second, a major finding of this work is that no single site is by itself enough to govern Cdc15 conformation, but rather that Cdc15 conformation depends mainly on the net charge. We have shown here and previously (Roberts-Galbraith et al., 2010) that Cdc15 phosphomutants of variable number of sites and position within the protein form condensates of various number and size in vivo that recruit other cytokinetic proteins and the effect correlates strictly with phosphorylation site number rather than localization on the protein.

2. In figure 4 the authors show that Cdc15-31A mutants prolong ring formation and decrease the time of ring constriction. They also claim that the time of ring maturation is decreased. However, it is well documented that altering the time of ring formation can inversely alter the time of ring maturation since ring maturation depends on the cells' ability to form a septum. A better way to describe the changes in the timing of cytokinetic events is to use the duplication of the spindle pole body marker as the starting event instead of using the completion of ring formation as a marker given that ring formation is itself impacted.

We apologize for the confusion regarding how we measured the kinetics of cytokinesis in our experiments. Duplication of the spindle pole body marker was used as the starting event to measure ring formation. We had initially measured this phase beginning with node formation as described, but these experiments were repeated using the SPB as marker, and we missed updating the text in line with the data. The data presented are correct, the text was not, and the text has been corrected.

Although there are instances in which the time of ring formation can inversely alter the time of ring maturation, this is not always the case. Please see Willet et al., 2015; Bhattacharjee et al., 2020 for two examples.

Reviewer #1 (Recommendations for the authors):Figure 1Some of the immunoblotting shown in this manuscript is not clear in demonstrating the phosphorylation status of Cdc15. Due to the technical limitation, it is understandable. Comparing the IP samples and lysate sample (only shown in B), I assume the lysate blot seems better to demonstrate the mobility shift (presumably due to dephosphorylation or degradation during IP?). Could the authors show F and G using the lysate samples to improve the quality of the blot?

The extent of gel separation of Cdc15 is variable but IPs are necessary for doing phosphatase treatments effectively. G is showing the contribution of Pck1, which is minor. We have replaced Figure 1F with a blot that has greater resolution of Cdc15 isoforms.

The mobility is less evident in the blots, so they should be enlarged in the figure panels to help readers. I recommend that this figure should only show the enlarged blot of analog-sensitive (AS) mutants, and the panels for temperature-sensitive or deletion mutants should be moved to supplemental figure 1.Panel (A) is the same experiment as the left two lanes of the panel (C), so this could be excluded from the figure or moved to supplemental figure 1.

We appreciate the reviewer recommendations but have elected to keep all the data together in one figure so that they can be directly compared. The described mobility shifts can be detected.

There are multiple bands in the blots. It is not clear which bands correspond to Cdc15/phosphor-Cdc15. Could the author indicate the target bands?The bottom bands are not shown correctly in panel (B) α-tubulin blot. The author should show which bands correspond to α-tubulin used as the loading control of the experiment. If the bottom bands correspond to α-tubulin, this blot has to be replaced with the one showing all the bands clearly.

We have indicated Cdc15 bands with brackets and what is α-tubulin with an arrowhead.

For the AS mutant experiments, it would be better to ensure that the kinase activity is inhibited. The authors may want to show the phosphorylation of a kinase substrate in the lysate blot as a lead-out of the kinase activity.

We are showing Cdc15, which is a known substrate of these kinases (Bhattacharjee et al., 2020; Lee et al., Curr Biol., 2018; Magliozzi et al., JCB, 2020).

Figure1-supplement 1This supplemental material seems not supplemental material for Figure 1. I assume this is supported information for Figure 2.

The reviewer is correct. We apologize for the mis-labeling and it has been corrected.

(A) In CB gel images, it is not clear which band corresponds to MBP/MBP-Cdc15N/MBP-Cdc15C. I recommend that the authors indicate them using arrowheads.

We elected not to add arrows to partA since the proteins in each reaction are already indicated above each lane.

(B) showed that the 19D mutant got less phosphorylation by Shk1/Pck1 than the 19A mutant. This suggests that Pom1-mediated phosphorylation inhibits Shk1/Pck1 to phosphorylate Cdc15. This supports the data shown in Figure 2, but the data is not discussed in the main text. The author should discuss the results if they include this result in this paper.

The objective of this experiment was to test whether there were sites of phosphorylation other than those targeted by Pom1 and this point was made with the A mutant. We are unsure how to interpret results with the D mutant that does not recapitulate phosphorylation (does not bind 14-3-3 protein Rad24, for example) and thus, have removed the use of the 19D mutant in these experiments.

Figure 2It may be helpful for readers to see schematics of the design of Cdc15C truncations.

A schematic has been added to Figure 2-supplement 1. In that figure, two different Cdc15 fragments were used. In Figure 2, only Cdc15C was used and it was defined in the text.

Figure 4I recommend that authors replace the bar graph with a swarm plot or violin plot, which gives readers variance in the data.

We have replaced the bar graphs with violin plots.

(A and B) The intensity of the Cdc15-mNG on the ring can be normalized by whole cell intensity.In the right panel of A, the intensity can be plotted against the ring diameter.(C) The same as A, the intensity of Rlc1-mCherry on the contractile ring can be normalized by whole cell intensity. For the right panel of C, the fluorescent intensity of the ring should be plotted against the ring diameter.

As suggested, these changes in data presentation have been made.

Figure 5I recommend that authors replace the bar graph with a swarm plot or violin plot, which gives readers variance in the data.

In this instance, the effects are so large that we elected to stay with bar graphs as visualization of the data.

Figure 7(F) It is unclear how authors quantify the fluorescent intensity of the edge and center of condensate. The author should show an example ROI for the measurement. I assume this was conducted by line plot or particle analysis.

We apologize for not explaining this thoroughly. The intensity was determined with a line scan and the approach has been detailed in the methods section.

(G and H) I assume the authors used His-tag and Ni-NTA interaction to anchor the His-IDR-SH3 on SLB. Although the BAR domain is not present in this construct, this must be clearly stated in the main text and figure legend to avoid confusion because Cdc15 is a well-known membrane-bound protein.

We have clarified the use of the His-tag and Ni-NTA lipids in the text and figure legend.

(G) should be His-Cdc15-IDR-SH3 rather than Cdc15-IDR-SH3.

We have added “His” to the labels of both G and H.

Figure 8(A) the bottom panel, the condensate shows the non-uniform distribution of the fluorescent, which is inconsistent with the author's conclusion from Figures 7 E and F. Does it because of the presence of other proteins? I see some patterns on the condensates.

It is very possible that some non-uniform distribution existed in some of these experiments using non-labeled components. We have re-done these experiments using different measures of protein concentration, as recommended by this reviewer, so the exact same images are no longer presented but yes, some discontinuity because of the unlabeled components is observed from time to time.

(B) The author stated that "we observed that 84% of droplets contained at least three" in line 361 in the main text. Could they show the ratio in the main figure? This should be discussed carefully in the manuscript. In my understanding, the author reconstitutes the Cdc15 condensation using equimolar protein concentration. It would be interesting to conduct the same experiment under the physiological molar ratio of these proteins.

Only 3 of the 4 proteins were labeled in this experiment which is why we can say that 84% contained at least 3 labeled proteins; they may contain 4. Also, the physiological molar ratios are debatable as available evidence on this is inconsistent. That said, we acknowledge that Cdc15 is likely to be in considerable excess at the ring so that all of these interactions could be supported in vivo simultaneously.

Figure 8 -supplemental 2I cannot see the images. All panels I have downloaded are black boxes.

We apologize for the confusion. Figure 8—figure supplement 2 contained all of the negative controls so they were indeed all black boxes containing no droplets. We have altered our figure presentation to include the negative controls alongside the positive results in new Figures 8 and 9.

Figure 9(A and B) The author should use a violin plot or swarm plot instead of the bar graph.

We think that the bar graph adequately describes the data and have elected not to change to a different display.

(C) could they also conduct FRAP in cells? Showing recovery could prove the LLPS rather than protein aggregation.

Unfortunately, we cannot conduct FRAP analysis due to the very small dimensions of the condensates that preclude analysis with available instrumentation.

DiscussionLine 485. One of the reference formats is not consistent with others.

We thank the reviewer for pointing this out. It has been corrected.

Materials and MethodRecombinant protein purificationLine 603 – 606The methodology of the protein concentration measurement should be shown individually. If the measurement is conducted in inconsistent ways, the author cannot prove the equimolar concentration in their experiments. And I have a little concern about their protein preparation. There are many degraded or contaminated proteins in the protein used in the experiments. In such a case, the concentration measurement using Nanodrop (absorbance of 280 nm light) may not be reliable.

We respectfully disagree. There are not many degraded or contaminating proteins used in our experiments (Please see the supplemental figures 1 for Figures 7-9). In the revised manuscript, we also repeated the experiments using protein concentrations determined by BCA and comparing to CB-stained proteins of known concentration. The results, now presented in Figures 7-9 are the same.

Quantification of mNG-Cdc15 PM to cytoplasmic ratioLine 695 – 697This section should have more description of the 2nd macro they used. The parameter from the 3D Object Counter plugin used for the calculation was not clearly stated in the section. In this way, it is difficult for the reader to understand and reproduce the experimental design in this study.

We apologize for not providing a more thorough description of the method. This has been corrected in the revised version with a detailed description in the methods section.

Reviewer #2 (Recommendations for the authors):This is a very well-written paper that provides a deep mechanistic understanding of how phosphorylation of intrinsically disordered regions can prevent phase separation and consequent protein function. This paper is very well suited for eLife. I have some suggestions mentioned below that will further strengthen the paper.1. As suggested above, phase separation of purified Cdc15-31A will provide strong evidence that the sites identified are physiologically involved in phase separation. Moreover, it would be useful to show if these mutants undergoing phase separation also recruit cytokinetic proteins.

This concern seems to be based on two misunderstandings. First, the method of producing and purifying phosphorylated Cdc15 was not well-described in the first version of the manuscript. This has been corrected. To elaborate, in order to purify Cdc15 from bacteria for LLPS assays, it must be co-produced with one of its kinases. Otherwise, it is not phosphorylated and it is not soluble. Thus, the proposed experiment of using recombinant Cdc15-31A is technically not possible. For the experiments presented in this paper, we co-produced Cdc15 with Pom1 to obtain the phosphorylated form of Cdc15. Pom1 phosphorylates 22 of the 31 sites mutated in the 31A mutant; these sites were identified in previous work and they are physiologically relevant in vivo (Bhattacharjee et al., 2020). Second, a major finding of this work is that no single site is by itself enough to govern Cdc15 conformation, but rather that Cdc15 conformation depends mainly on the net charge. We have shown here and previously (Roberts-Galbraith et al., 2010) that Cdc15 phosphomutants of variable number of sites and position within the protein form condensates of various number and size in vivo that recruit other cytokinetic proteins and the effect correlates strictly with phosphorylation site number rather than localization on the protein.

2. In figure 4 the authors show that Cdc15-31A mutants prolong ring formation and decrease the time of ring constriction. They also claim that the time of ring maturation is decreased. However, it is well documented that altering the time of ring formation can inversely alter the time of ring maturation since ring maturation depends on the cells' ability to form a septum. A better way to describe the changes in the timing of cytokinetic events is to use the duplication of the spindle pole body marker as the starting event instead of using the completion of ring formation as a marker given that ring formation is itself impacted.

Please see responses above.

3. Also, Figure 8—figure supplement 2 is very hard to follow. The figure is too dark and the signal contrast is barely visible but upon zooming in, I was able to see dots. Inverting the image would help with the lack of contrast. This figure also misses a scale bar.

We apologize for the confusion. Figure 8—figure supplement 2 contained all of the negative controls so they are indeed all blank. We have altered our figure presentation to include the negative controls alongside the positive results in new Figures 8 and 9.